# A Learning-Augmented Approach
# to Online Allocation Problems

**Ilan Reuven Cohen**
Faculty of Engineering
Bar-Ilan University
Ramat Gan, Israel.
ilan-reuven.cohen@biu.ac.il

**Debmalya Panigrahi**
Department of Computer Science
Duke University
Durham, NC, USA.
debmalya@cs.duke.edu

## Abstract

In online allocation problems, an algorithm must choose from a set of options at each step, where each option incurs a set of costs/rewards associated with a set of $d$ agents. The goal is to minimize/maximize a function of the accumulated costs/rewards assigned to the agents over the course of the entire allocation process. Such problems are common in combinatorial optimization, including minimization problems such as machine scheduling and network routing, as well as maximization problems such as fair allocation for welfare maximization.

In this paper, we develop a general learning-augmented algorithmic framework for online allocation problems that produces a nearly optimal solution using only a single $d$-dimensional vector of learned weights. Using this general framework, we derive learning-augmented online algorithms for a broad range of application problems in routing, scheduling, and fair allocation. Our main tool is convex programming duality, which may also have further implications for learning-augmented algorithms in the future.

## 1 Introduction

In the last few years, tremendous advances in machine learning have triggered much interest and rapid progress in a new domain called *learning-augmented* online algorithms. In many practical scenarios, one needs to solve optimization problems with dynamically evolving requirements (called online algorithms), where uncertainty about the future significantly affects the quality of the solution. The basic premise of the learning-augmented paradigm is to use (possibly noisy) machine-learned predictions as a proxy for the future, and design online algorithms that can take advantage of good predictions while not falling prey to bad ones. (The resulting algorithms are sometimes also referred to as "online algorithms with predictions".) The explosive growth of this new area over the last few years has spanned a large variety of domains such as caching, scheduling, matching, clustering, network design, and many others (see related work), and generated much excitement at the intersection of algorithm design and machine learning. While many novel and clever algorithms have been designed in the learning-augmented setting, these are typically tailored to the requirements of a specific problem at hand, and do not generalize to broader problem classes. Indeed, there is a surprising absence of unifying algorithmic models and general-purpose tools for online algorithms with predictions, particularly when contrasted with other models of computation where algorithmic progress has largely relied on broad, unifying models and algorithmic techniques.

In this paper, we aim to partially rectify this situation by studying a broad class of problems called online allocation in the learning-augmented setting. The setup is the following: in each online step, the algorithm is presented a set of options and has to choose one of them. Each option is represented by a vector, and the final objective is a (linear or non-linear) function of the (coordinate-wise) sum

39th Conference on Neural Information Processing Systems (NeurIPS 2025).

of all the chosen vectors. As we will soon see, this captures a broad range of problems in network routing, scheduling, fair allocation, etc. In this paper, we develop a general algorithmic strategy for this entire class of problems in the learning-augmented setting that obtains a nearly optimal (fractional) solution. This is in sharp contrast to strong (super-constant) lower bounds without ML advice, for almost all problems captured by this framework.

The online allocation problem that we described above has many applications. Among the most important is that of online routing. In this problem, we are given a network and in each online step, the algorithm has to choose a route from a given source to a destination node. Eventually, the goal is to minimize the maximum congestion of any network link, which is defined as the number of chosen routes using that link. This is a classical and well-studied problem in online algorithms, and it is well-known that the best competitive ratio[1] achievable is logarithmic in the size of the network [3] (see also [4, 6, 7]). In sharp contrast, we obtain a $(1 + \epsilon)$-approximation for any $\epsilon > 0$ using learning augmentation, by modeling the problem as online allocation where each coordinate in the vector represents a network link and the goal is to minimize the $\ell_\infty$-norm of the congestion vector.

A second application domain for online allocation is that of online scheduling problems. Consider the problem of processing jobs online where each job comes with a slate of options given by vectors that represent the processing time on different machines. Overall, the goal is to minimize the makespan, i.e., the maximum load on any machine. This problem is also described via online allocation, where the coordinates of the vectors represent machines and the objective is again the $\ell_\infty$-norm of machine loads given by the sum of the selected vectors. Our online allocation algorithm gives a $(1 + \epsilon)$-competitive learning-augmented solution to this problem, which again sharply contrasts with lower bounds that are logarithmic in the number of machines in the absence of learning augmentation [10].

We note that the flexibility of the online allocation framework also allows us to extend these problem formulations immediately to a much broader set of objectives, such as all $\ell_p$-norm objectives [5] rather than just the $\ell_\infty$-norm. Another interesting direction is that of maximization problems. In contrast to minimization problems where the vector coordinates can be thought of as resources whose use needs to be minimized, we now imagine the coordinates to represent value added to agents when a particular option is selected. This allows us to model fair allocation problems, where a set of items have to be assigned in a way that maximizes the minimum value along all agents (or another function such as Nash social welfare, $p$-means, etc.) [13, 15] Again, we observe the sharp contrast between the traditional online setting without learning augmentation, where strong polynomial lower bounds exist for these problems [34, 18] and the learning-augmented setting where our online allocation algorithm immediately yields a $(1 - \epsilon)$-approximation for any $\epsilon > 0$.

We end this section by commenting on the machine-learned parameters that we use in our learning-augmented algorithm for the online allocation problem. As described above, we show that there exists a set of learned parameters that allows us to approximate the optimal objective within a $(1 \pm \epsilon)$ factor of the optimal value. Importantly, we show that these parameters are bounded as a function of $\epsilon$ and the number of coordinates (rather than the number of steps or options) and, moreover, that our learning-augmented algorithm is resilient to small errors in the values of the learned parameters. This allows us to derive PAC-learning results for the learned parameters and to use them in the learning-augmented algorithm to obtain a $(1 \pm O(\epsilon))$-approximation.

**Organization.** The formal model and statement of results appear in Section 2. Section 3 presents related work. The main result, which provides the learning-augmented algorithm, appears in Section 4. Applications of this result to routing and fair allocation problems are discussed in Section 5. In Appendix C, we derive sample complexity bounds for the learned parameters in the PAC model. All omitted details of proofs are in the appendix.

## 2 Our Contributions

First, we define the online allocation framework:

**Online Allocation Problem.** In this problem, we receive an initial input of a function $f$ over a bounded domain in $\mathbb{R}_+^d$. Subsequently, we proceed in steps. At every time step $t = 1, \ldots, T$, we receive a set $A_t \subseteq \mathbb{R}_+^d$ of $d$-dimensional vectors. We must select one vector $\mathbf{v}_t^\dagger \in A_t$ before

---

[1]As is usual in online algorithms, performance of an algorithm is measured via its competitive ratio, which is the worst-case ratio of the algorithm's objective to that of an optimal solution.

proceeding to time step $t + 1$, using only information available up to time $t$. Denote the result vector $\mathbf{v}_{\text{tot}}^{\dagger} := \sum_{t=1}^{T} \mathbf{v}_t^{\dagger}$. The goal is to maximize or minimize $f(\mathbf{v}_{\text{tot}}^{\dagger})$ for some objective function $f(\cdot)$.

For ease of notation, we fix an arbitrary ordering of the vectors in $A_t$, and denote by $v_{t,k} \in \mathbb{R}_+^d$ the $k$th vector in $A_t$, i.e., $A_t = \langle v_{t,1}, \ldots, v_{t,|A_t|} \rangle$. Accordingly, we define $K(t) = \{1, \ldots, |A_t|\}$ as the set of indices of these vectors, and for each $k \in K(t)$, we let $v_{i,t,k}$ denote the $i$th coordinate of vector $v_{t,k}$ in $A_t$. Our results apply to *well-behaved functions* $f(\cdot)$ that we define as follows:

**Well-Behaved Functions.** Let $f : \mathbb{R}_+^d \to \mathbb{R}_+$ be the objective function defined on the result vector. Then, $f$ is well-behaved if it satisfies the following properties:

*Monotonicity:* $f$ is said to be *monotone* if for any $\ell, \ell' \in \mathbb{R}_+^d$ such that $\ell_i \geq \ell_i'$ for all $i \in [d]$, we have $f(\ell) \geq f(\ell')$.

*Homogeneity:* $f$ is said to be *homogeneous* if for any $\ell, \ell' \in \mathbb{R}_+^d$ such that $\ell_i' = \alpha \cdot \ell_i$ for all $i \in [d]$, we have $f(\ell') = \alpha \cdot f(\ell)$.

These properties are satisfied by most objective functions studied in the literature including linear functions, $p$-norms, Nash Social Welfare (which is the geometric mean), $p$-means, etc.

Our algorithmic scheme will focus on developing a fractional selection rule, which can be interpreted as a probability distribution over the different choices in every step. We formally describe a fractional solution as follows:

**Fractional Solution.** At step $t$, the algorithm has to assign a fractional value to each option $x_{t,k} \in [0, 1]$ for $k \in K(t)$ such that $\sum_{k=1}^{|K(t)|} x_{t,k} = 1$. Accordingly, we have

$$\mathbf{v}_{\text{tot}}^f := \sum_{t=1}^{T} \sum_{k=1}^{|A_t|} \mathbf{v}_{t,k} \cdot x_{t,k}.$$

Note that in some scenarios such as the allocation of divisible items or network flow routing, a fractional solution is already sufficient.

**Learning-Based Online Scheme for Allocation Problems.** Our main result is that for any $d$-dimensional instance of the online allocation problem with a well-behaved objective function, there exists a vector $\alpha \in \mathbb{R}^d$ such that an online algorithm guided by $\alpha$ achieves a competitive ratio of $1 + \epsilon$ for minimization or $1 - \epsilon$ for maximization, for any $\epsilon > 0$.

**Theorem 2.1.** *Given an instance of the online allocation problem with a well-behaved objective and any $\epsilon > 0$, there exists a set of learned parameters $\alpha \in \mathbb{R}^d$ and an online algorithm that uses $\alpha$ such that the resulting fractional solution achieves a $(1 \pm O(\epsilon))$-approximation.*

The online algorithm that establishes this result uses a simple *exponential assignment rule*, where given a learned parameter vector $\alpha \in \mathbb{R}^d$, the fractional allocation $x_{tk}(\alpha)$ of option $k \in K(t)$ is:

$$x_{tk}(\alpha) \propto \exp \left( -\sum_{i=1}^{d} \alpha_i \cdot \frac{v_{i,t,k}}{\mathbf{m}_t(v)} \right), \text{ where } \mathbf{m}_t(v) = \min\{v_{itk} | v_{itk} > 0, i \in [d], k \in K(t)\}.$$

We further show that the learned parameters can be bounded as a function of $d$ and $\epsilon$, and that the algorithm is robust to small perturbations in the parameter values. This structure allows the parameter space to be both bounded and discretized in terms of $d, \epsilon$, independent of the time horizon. This property is crucial in applications where the number of steps (which may represent flow requests, jobs, or items) is significantly larger than the number of dimensions (which may represent network links, machines, or agents). Moreover, the discretization of the parameter space ensures that the parameters are learnable, and allows us to bound the number of samples required for learning.

**Learnability of Parameters.** Following the PAC framework of [26], we establish the learnability of parameter vectors for well-behaved objectives, under the additional assumption of *superadditivity* for maximization or *subadditivity* for minimization, as introduced in [18].

We consider a setting in which each online step (i.e., a set of vectors) is drawn independently (though not necessarily identically) from a distribution, and each step contributes only a small fraction of the total allocation. This latter property is formalized by the standard *small items assumption* (see, e.g., [1, 26, 24, 29]), which we state below:

*Small Items Assumption:* There exists a $\zeta = \Theta\left(\frac{\log d}{\epsilon^2}\right)$ such that $v_{itk} \leq \frac{L}{\zeta}$ for every $i \in [d], t \in [T]$, and $k \in K(t)$.

Under this assumption, we derive a sample complexity bound for approximately learning a parameter vector $\alpha$ that ensures near-optimal performance. The following informal theorem summarizes our main learnability result; see Appendix C for details:

**Theorem 2.2.** *Under standard PAC assumptions, the small items assumption, and mild regularity conditions on the objective (subadditivity for minimization and superadditivity for maximization), the parameter vector $\alpha \in \mathbb{R}^d$ can be learned from $O\left(\frac{d}{\log d} \cdot \log\frac{d}{\epsilon}\right)$ i.i.d. samples.*

**Rounding the Fractional Solution.** As mentioned earlier, a fractional solution is already sufficient for many applications. Furthermore, we show that under the small items assumption (see above), a fractional solution can be converted into a randomized online integral solution with a $(1 \pm \epsilon)$ loss in the objective value, where the parameter $\epsilon$ only depends on the parameters of the small items assumption.

Consider the following online randomized rounding algorithm: given an online fractional solution $x_{tk}$ for $t \in [T]$, the algorithm chooses option $k$ with probability $x_{t,k}$. Under the small items assumption, standard Chernoff bounds ensure that with high probability, the maximum coordinate in the rounded vector remains within a multiplicative factor of $(1 \pm O(\epsilon))$ of its fractional counterpart. We get the following theorem:

**Theorem 2.3.** *Let $\mathbf{v}_{tot}^f$ be a fractional solution, and let $\mathbf{v}_{tot}^\dagger$ be the integer solution produced by randomized rounding of this fractional solution. Then, under the small items assumption, the following holds with high probability:*

$$\max_{i \in [d]} \mathbf{v}_{tot,i}^\dagger \leq (1 + \epsilon) \cdot \max_{i \in [d]} \mathbf{v}_{tot,i}^f, \text{ and}$$
$$\min_{i \in [d]} \mathbf{v}_{tot,i}^\dagger \geq (1 - \epsilon) \cdot \min_{i \in [d]} \mathbf{v}_{tot,i}^f.$$

Now, combining the above theorem with Theorem 2.1, we get the following result for integer solutions under the small items assumption:

**Corollary 2.4.** *Given an instance of the online allocation problem with a well-behaved objective and under the small-item assumption, for any $\epsilon > 0$, there exists a set of learned parameters $\alpha \in \mathbb{R}^d$ and an online algorithm that uses $\alpha$ such that the resulting integral solution achieves a $(1 \pm O(\epsilon))$-approximation.*

**Robustness–Consistency Tradeoff.** Ideally, a learning-augmented algorithm should simultaneously exploit accurate predictions to achieve near-optimal performance (consistency), while also maintaining strong worst-case guarantees when predictions are unreliable (robustness). The (not learning-augmented) worst case lower bounds for allocation problems with minimization (resp., maximization) objectives is $\Omega(\log d)$ (resp., $\Omega(d)$). Formally, an algorithm is said to be $\gamma$-*consistent* and $\delta$-*robust* if it achieves a $\gamma$-approximation under accurate predictions (consistency), and a $\delta$-approximation in the worst case when predictions are unreliable (robustness). In Appendix D, we give a slight modification of our scheme that preserves robustness while achieving consistency for both minimization and maximization objectives.

**Theorem 2.5.** *For minimization objectives, there exists an algorithm that uses $\alpha \in \mathbb{R}^d$, a predicted parameter vector, and achieves $1$-consistent and $O(\log d)$-robust approximation.*

*For maximization objectives, there exists an algorithm that uses $\alpha \in \mathbb{R}^d$, a predicted parameter vector, and a parameter $\lambda$, which achieves a $(1 - \lambda)$-consistent and $\lambda \cdot d$-robust approximation.*

**Applications.** In Section 5, we illustrate the utility of our framework in handling a broad class of objectives by demonstrating its use in two applications. The first is for the online routing problem, where we are given a network and in each time-step, there is request to route a given value of flow from a source to a destination vertex [3]. The flow needs to be routed in a way that minimizes maximum congestion on any edge cumulatively across all time-steps. We give the first learning-augmented algorithm for this problem as a simple corollary of our general online allocation framework. Next,

we consider a maximization problem, that of allocating items to agents so as to maximize the Nash Social Welfare [13]. Again, we give a learning-augmented algorithm for this problem as a simple corollary of our general framework, matching previous results in [18]. We note that our framework applies to many other application problems in domains such as scheduling (makespan and $\ell_p$-norm minimization) and fair allocation (Santa Claus and $p$-means maximization) that we do not state here for brevity.

## 3   Related Work

Previous work that is mostly closely related to ours is on learning-augmented online scheduling and assignment problems [23, 24, 26, 18]. These works focus on the assignment of items that arrive online to agents, where each choice affects only a single agent. In our setting, this corresponds to each choice being represented by a $d$-dimensional vector with a single nonzero dimension, where $d$ corresponds to the number of agents.

Lattanzi et al. [23] focused on minimizing the makespan (maximum load) for restricted assignment. They showed that a suitable set of learned parameters enables a proportional allocation rule that obtains a nearly optimal result. In [24], it was further shown that these parameters are PAC-learnable.

Li and Xian [26] generalized this framework to handle unrelated scheduling. Their scheme introduces two learned parameters per machine. The first set of parameters transform the instance into a restricted-related setting, and they show that proportional assignment using the second set of parameters then applies to this restricted setting.

Cohen and Panigrahi [18] improved this result by using only a single set of parameters in an exponential allocation scheme, and also extended it to handle a large number of maximization and minimization objectives.

Our scheme generalizes these previous works to accommodate general vectors, which is crucial for applications such as online routing that cannot be solved using the previous techniques. Furthermore, we simplify the previous approaches as well by requiring only a single variable per dimension, thereby eliminating the dependence on an additional exponent base which further simplifies the analysis of the existence of such parameters. Our work introduces a new set of techniques based on perturbation/sensitivity analysis because the ideas previously used for bounding the learned parameters do not generalize to the case of arbitrary vector options.

In [14], the authors study the Online Nash Social Welfare problem with predictions. The goal is to compute an online divisible allocation of goods among $d$ agents in a way that balances fairness and efficiency. In their setting, the prediction for each agent is their *monopolist value*—the utility the agent would obtain if all resources were allocated solely to them—and their algorithm achieves an $O(\log d)$-approximation under this assumption. In contrast, we show that by learning a different $d$-dimensional parameter vector, it is possible to achieve a nearly optimal allocation.

Various papers [12, 35, 19, 20] investigate the robustness–consistency tradeoff in the context of general packing and covering problems. These algorithms typically introduce an additional parameter $\lambda \in [0, 1]$ to encode the algorithm's confidence in the prediction. A primal-dual scheme is then employed to interpolate between a worst-case baseline and a prediction-driven strategy. Unlike our approach, these methods assume that the prediction consists of the full solution, which is often impractical in real-world settings.

More broadly, the study of learning-augmented online algorithms was initiated by Lykouris and Vassilvitskii [27] in the context of the caching problem and has since grown into a prominent research area. This framework enhances online algorithms by incorporating machine-learned predictions about the future, allowing them to surpass pessimistic worst-case competitive bounds. Over the past few years, numerous online allocation problems have been explored within this framework, including applications in scheduling [33, 8, 9, 11, 21, 30], online matching [2, 17, 22], and ad delivery [28, 25]. For a broader overview of learning-augmented online algorithms, we refer the reader to the surveys by Mitzenmacher and Vassilvitskii [31, 32].

# 4 Online Allocation for a Well-Behaved Objective via Learned Parameters

In this section, we show that for the general case of the online allocation problem, there exists a set of learned parameters that guarantees a near-optimal solution. Specifically, we show the following theorem, which is a more refined version of (and establishes) Theorem 2.1:

**Theorem 4.1.** *Given an instance of the online allocation problem with a well-behaved objective and any $\epsilon > 0$, there exists a set of learned parameters $\alpha \in \textbf{NET}(q, s)$ and an online algorithm that uses $\alpha$ such that the resulting fractional solution achieves a $(1 \pm O(\epsilon))$-approximation.*

Here, $\textbf{NET}(q, s) = \left\{ \frac{i}{S} \mid i \in [0, q \cdot s] \right\}^d$ denotes a $d$-dimensional discrete net with parameters $q, s$ that are bounded in $\text{poly}(d, 1/\epsilon)$.

For the minimization of a well-behaved function, we first consider the **MinMax** objective, which is defined as the minimization of $f$, where $f(v) = \max_{i \in [d]} v_i$. Similarly, for the maximization of a well-behaved function, we first consider the **MaxMin** objective, which is defined as the maximization of $f$, where $f(v) = \min_{i \in [d]} v_i$.

In this section, we focus on the **MinMax** objective. In Appendix E, we address the **MaxMin** objective. In Appendix F, we explain how to extend these results to general *well-behaved* objectives.

## 4.1 Learning-Augmented Online Allocation for the MinMax Objective

We now state the main result for the **MinMax** objective:

**Theorem 4.2.** *Given an instance of the online allocation problem with a **MinMax** objective and any $\epsilon > 0$, there exists a set of learned parameters $\alpha \in \textbf{NET} \left( \frac{d^2}{\epsilon} \cdot \ln \left( \frac{d}{\epsilon} \right), \frac{d^3}{\epsilon^3} \right)$ and an online algorithm that uses $\alpha$ such that the resulting fractional solution is a $(1 + O(\epsilon))$-approximation.*

To prove this result, we first apply a pre-processing step to convert arbitrary instances of the problem to structured instances that we say are *balanced*. Note that the conversion to balanced instances is an algorithmic technique and not a restriction on the input. We then leverage convex programming duality on a max-entropy style convex programming formulation to show the existence of learned parameters that can be used to obtain an near-optimal solution online. At this stage, the learned parameters can be of arbitrary precision and are not necessarily efficiently learnable. To ensure the latter, we need to obtain parameters that belong to $\textbf{NET} \left( \frac{d^2}{\epsilon} \cdot \ln \left( \frac{d}{\epsilon} \right), \frac{d^3}{\epsilon^3} \right)$. We do this in the last step by using tools from perturbation and sensitivity analysis in the convex programming literature. Together, these steps establish Theorem 4.2.

**Preprocessing of MinMax Instances.** Recall that the exponential assignment scheme computes online allocations based on the ratio $\frac{v_{i,t,k}}{\mathbf{m}_t(v)}$. This ratio can be highly sensitive and potentially unbounded under small perturbations to the input, thereby making it impossible to execute the last step of designing learned parameters of bounded precision. To overcome this difficulty, we introduce a pre-processing step that transforms the original instance into a *balanced* instance, incurring only a small loss in the objective value.

An instance $\tilde{I}(\tilde{v}, \tilde{K})$ is said to be *balanced* if it satisfies the following condition for all $t \in [T]$, $i \in [d]$, and $k \in K(t)$:

$$\frac{\tilde{v}_{itk}}{\mathbf{m}_t(\tilde{v})} \in \{0\} \cup \left\{ (1+\epsilon)^b \,\middle|\, b \in \mathbb{Z}_+ \cap \left[0, \log_{1+\epsilon} \left( \frac{d^2}{\epsilon^2} \right) \right] \right\}.$$

Note that in a balanced instance, we have $\frac{v_{i,t,k}}{\mathbf{m}_t(v)} \leq \frac{d^2}{\epsilon^2}$. Moreover, by omitting redundant options, we may assume that the number of options per step is bounded, i.e., $\ln |K(t)| = O\left(d \cdot \ln(d/\epsilon)\right)$. We construct a balanced instance $I(\tilde{v}, \tilde{K})$ as follows:

Given an instance $I(v, K)$,

    1. Define $\hat{v}$ such that

$$\hat{v}_{itk} = \begin{cases} 0 & \text{if } \frac{v_{itk}}{\max_{i'} v_{i'tk}} < \frac{\epsilon}{d} \\ v_{itk} & \text{otherwise.} \end{cases}$$

2. Define $\tilde{K}$ by removing option $k'$ from step $t$ whenever there exists another option $k$ with

$$\max_i \hat{v}_{i,t,k'} > \frac{d}{\epsilon} \cdot \max_i \hat{v}_{i,t,k}.$$

3. Define $\tilde{v}$ as follows: if $\hat{v}_{itk} = 0$, then set $\tilde{v}_{itk} = 0$; otherwise, set

$$\tilde{v}_{itk} = \mathbf{m}_t(\hat{v}) \cdot (1 + \epsilon)^{\left\lfloor \log_{1+\epsilon} \left( \frac{\hat{v}_{itk}}{\mathbf{m}_t(\hat{v})} \right) \right\rfloor}.$$

By definition, $I(\tilde{v}, \tilde{K})$ is a balanced instance. In Appendix A, we show that transforming $I(v, K)$ into $I(\tilde{v}, \tilde{K})$ incurs only a $(1 + O(\epsilon))$ loss in the objective value.

**Lemma 4.3.** *Let $I(v, K)$ be an instance of the allocation problem with the **MinMax** objective, and let $\epsilon > 0$. Then, any $(1 + \epsilon)$-approximate solution to the balanced instance $I(\tilde{v}, \tilde{K})$ constructed via the pre-processing algorithm yields a $(1 + O(\epsilon))$-approximate solution to $I(v, K)$.*

**Learned Parameters for Balanced Instances.** In light of Lemma 4.3, it suffices to only consider balanced instances in Theorem 4.2. We restate this goal as follows:

**Lemma 4.4.** *Given a balanced instance of the online allocation problem with a **MinMax** objective and $\epsilon > 0$, there exists a set of parameters $\alpha \in \mathbf{NET}\left( \frac{d^2}{\epsilon} \cdot \ln\left(\frac{d}{\epsilon}\right), \frac{d^3}{\epsilon^3} \right)$ such that the fractional solution defined by the exponential assignment scheme with parameters $\alpha$ is a $(1 + O(\epsilon))$-approximation to the optimal objective.*

The remainder of this section gives a proof of Lemma 4.4. We do this in two steps. First, we prove the existence of parameters $\alpha^{(\epsilon)}$ that are bounded as a function of $d, \epsilon$ using which the exponential assignment scheme produces an $(1 + \epsilon)$-approximate optimal solution. (For the special case of $\epsilon = 0$, the solution is precisely optimal but the parameters may be unbounded.) Then, we show that small errors (up to $\epsilon^3/d^3$) in the parameter values incur only a $(1 + O(\epsilon))$ loss in the objective value, enabling discretization of the learned parameters to $\mathbf{NET}\left( \frac{d^2}{\epsilon} \cdot \ln\left(\frac{d}{\epsilon}\right), \frac{d^3}{\epsilon^3} \right)$.

We use $L^*$ to denote the optimal **MinMax** value, and define the following convex program for $\epsilon > 0$:

$$
\begin{aligned}
\min \quad & \sum_{t \in [T]} \mathbf{m}_t(v) \sum_{k \in K(t)} x_{tk} \cdot (\ln x_{tk} - 1) \\
\text{s.t.} \quad & \sum_{t \in [T]} \sum_{k \in K(t)} v_{itk} \cdot x_{tk} \leq L^* \cdot (1 + \epsilon), && \forall i \in [d], \\
& \sum_{k \in K(t)} x_{tk} = 1, && \forall t \in [T], \\
& x_{tk} \geq 0, && \forall k \in K(t), t \in [T]
\end{aligned}
$$

Figure 1: Convex Programming Formulation for the **MinMax** Objective

**Lemma 4.5.** *Given an instance of the online allocation problem with the **MinMax** objective and any $\epsilon \geq 0$, there exists a vector $\alpha^{(\epsilon)} \in \mathbb{R}^d_+$ such that the fractional solution defined by the exponential assignment scheme with parameters $\alpha^{(\epsilon)}$ is $(1 + O(\epsilon))$-approximately optimal.*

*Proof.* Given such instance and for fixed $\epsilon$ consider the convex program of Figure 1. By our assumption, $L^*$ is the optimal **MinMax** objective therefore there exists a feasible solution for the convex program for any $\epsilon \geq 0$. Accordingly, define the Lagrangian $L(x, \alpha, \beta)$ as

$$\sum_{t \in [T]} \mathbf{m}_t(v) \sum_{k \in K(t)} x_{tk} \ln\left(\frac{x_{tk}}{e}\right) + \sum_{i \in [d]} \alpha_i \left( \sum_{t \in [T]} \sum_{k \in K(t)} v_{itk} \cdot x_{tk} - L^*(1+\epsilon) \right) + \sum_{t \in [T]} \beta_t \left( 1 - \sum_{k \in K(t)} x_{tk} \right).$$

From the KKT conditions for the optimal solution to the convex program as a function of $\epsilon$ $x^{(\epsilon)}, \alpha^{(\epsilon)}, \beta^{(\epsilon)}$, the solution that allocates according to $x^{(\epsilon)}$ is a $(1 + \epsilon)$-approximation to the optimal

objective $L^*$, and $\alpha_i^{(\epsilon)} \geq 0$ for all $i \in [d]$. Furthermore,

$$\frac{dL}{dx_{tk}} = 0 \quad \text{for all } k \in K(t), \text{ which gives } \mathbf{m}_t(v) \cdot \ln(x_{tk}^{(\epsilon)}) + \mathbf{m}_t(v) \cdot \sum_i \alpha_i^{(\epsilon)} \cdot v_{itk} = \beta_t^{(\epsilon)}.$$

For any two options $k, r \in K(t)$, we obtain:

$$\mathbf{m}_t(v) \cdot \ln(x_{tk}^{(\epsilon)}) + \sum_i \alpha_i^{(\epsilon)} \cdot v_{itk} = \mathbf{m}_t(v) \cdot \ln(x_{tr}^{(\epsilon)}) + \sum_i \alpha_i^{(\epsilon)} \cdot v_{itr}.$$

Therefore, $\ln\left(\dfrac{x_{tk}^{(\epsilon)}}{x_{tr}^{(\epsilon)}}\right) = \sum_i \alpha_i^{(\epsilon)} \cdot \dfrac{v_{itr}}{\mathbf{m}_t(v)} - \sum_i \alpha_i^{(\epsilon)} \cdot \dfrac{v_{itk}}{\mathbf{m}_t(v)}$. Coupled with $\sum_{k \in K(t)} x_{tk}^{(\epsilon)} = 1$, we get

$$x_{tk}^{(\epsilon)} \propto \exp\left(-\sum_i \alpha_i^{(\epsilon)} \cdot \frac{v_{itk}}{\mathbf{m}_t(v)}\right). \qquad \square$$

**Bounding the Learned Parameters.** For $\epsilon = 0$, the learned parameters $\alpha^{(\epsilon)}$ in the previous lemma may be unbounded. However, for $\epsilon > 0$, we show that each $\alpha_i^{(\epsilon)}$ can be bounded as a function of $d$ and $\epsilon$. Our main tool is perturbation and sensitivity analysis, following the framework of Boyd et al. [16]. (See Appendix B for details of perturbation and sensitivity analysis.)

**Lemma 4.6.** *Let $x^{(\epsilon)}, \alpha^{(\epsilon)}, \beta^{(\epsilon)}$ be the optimal solution to the convex program in Figure 1, for some $\epsilon > 0$. Then, for all $i \in [d]$, it holds that $\alpha_i^{(\epsilon)} \leq \frac{d^2}{\epsilon} \cdot \ln\left(\frac{d}{\epsilon}\right)$.*

*Proof.* We need the following claim:

**Claim 4.7.** *For an $n$ dimensional vector $x \in [0,1]^n$ such that $\sum_{i=1}^n x_i = 1$ we have $\sum_{k=1}^n x_k \ln x_k \geq -\ln n$.*

*Proof.* By Jensen's inequality, we have

$$\varphi\left(\sum_i p_i \cdot y_i\right) \geq \sum_i p_i \cdot \varphi(y_i), \text{ where } p_i \geq 0, \sum p_i = 1, \text{ and } \varphi \text{ is concave.}$$

The desired bound now follows by setting $\varphi(x) = \ln(x)$, $p_i = x_i$ and $y_i = 1/x_i$. $\qquad \square$

We define a perturbed convex program based on Figure 1, where $u_i$ corresponds to the constraint $\alpha_i$. For each $i \in [d]$, setting $u_i = -\epsilon \cdot L^*$ and $u_{i'} = 0$ for $i' \neq i$ ensures that constraint $i$ in the perturbed problem matches the original constraint, thereby guaranteeing a feasible solution. By Lemma B.1,

$$p^*(0,0) \geq p^*(u,v) + \alpha_i^{(\epsilon)} \cdot \epsilon \cdot L^*, \text{ which implies}$$

$$\alpha_i^{(\epsilon)} \cdot \epsilon \cdot L^* \leq p^*(0,0) - p^*(u,v) \leq \sum_{t \in [T]} \mathbf{m}_t(v) \ln |K(t)|,$$

where the last inequality follows from Claim 4.7.

$$\sum_t \mathbf{m}_t(v) = \sum_t \mathbf{m}_t(v) \sum_k x_{tk}^* \leq \sum_t \sum_k x_{tk}^* \sum_i v_{itk} = \sum_i \sum_t \sum_k v_{itk} \cdot x_{tk}^* \leq \sum_i L^* = d \cdot L^*,$$

where the first equality holds because $x^*$ is a feasible solution, the first inequality follows since there must be at least one nonzero coordinate and by definition of $\mathbf{m}_t(v)$, and the second inequality follows from the linear program constraint for dimension $i$.

By our assumption on the instance, we have

$$\ln |K(t)| \leq d \cdot \ln\left(\log_{1+\epsilon}\left(\frac{d^2}{\epsilon^2}\right)\right) \leq d \cdot \ln\left(\frac{d}{\epsilon}\right). \text{ Therefore, } \alpha_i^{(\epsilon)} \leq \frac{d^2}{\epsilon} \cdot \ln\left(\frac{d}{\epsilon}\right). \qquad \square$$

**Discretizing the Learned Parameters and Noise Resilience.** In order to prove Lemma 4.4, we need to ensure that the learned parameters $\alpha^{(\epsilon)}$ belong to the discrete set $\textbf{NET}\left(\frac{d^2}{\epsilon} \cdot \ln\left(\frac{d}{\epsilon}\right), \frac{d^3}{\epsilon^3}\right)$. This may not be true of the Lagrangian multipliers derived above. We show that in discretizing the learned parameters, the competitive ratio only worsens by $(1 + O(\epsilon))$. In particular, if we replace the Lagrangian vector $\alpha^{(\epsilon)}$ with a perturbed vector $\tilde{\alpha}$, where $-\frac{\epsilon^3}{d^3} \leq \tilde{\alpha}_i - \alpha_i^{(\epsilon)} \leq \frac{\epsilon^3}{d^3}$ for all $i \in [d]$, then the assignment fractions change by at most $1 \pm 4\epsilon$. Note that this lemma is also important from a noise resilience perspective: if the learned parameters have small error, this lemma shows that the resulting allocation is still approximately optimal.

**Lemma 4.8.** *Let $\alpha^* \in \mathbb{R}^d$ be a vector, and let $\tilde{\alpha} \in \mathbb{R}^d$ be a perturbed vector such that $|\tilde{\alpha}_i - \alpha_i^*| \leq \frac{\epsilon^3}{d^3}$ for all $i \in [d]$. Then, the fractional assignment $x_{tk}(\tilde{\alpha})$ satisfies*

$$(1 - 4\epsilon)\, x_{tk}(\alpha^*) \leq x_{tk}(\tilde{\alpha}) \leq (1 + 4\epsilon)\, x_{tk}(\alpha^*) \quad \text{for all } k \in K(t), t \in [T].$$

*Proof.* We need the following claim:

**Claim 4.9.** *For any $\epsilon > 0$, if we are given two sets of $K$ weights $a, a' \in \mathbb{R}_+^K$ such that $1 - \epsilon \leq \frac{a_k}{a_k'} \leq 1 + \epsilon$, then for $x, x' \in [0,1]^K$ such that $\sum_{k=1}^K x_k = \sum_{k=1}^K x_k' = 1$ and $x_k \propto a_k$ and $x_k' \propto a_k'$, we have $1 - 4\epsilon \leq \frac{x_k}{x_k'} \leq 1 + 4\epsilon$.*

*Proof.* Consider the assignment fractions $x_k$. By definition, we have the following:

$$x_k = \left(\sum_{r=1}^K \frac{a_r}{a_k}\right)^{-1} \leq \left(\sum_{r=1}^K \frac{a_r'}{a_k'} \cdot \frac{1-\epsilon}{1+\epsilon}\right)^{-1} \leq \frac{1+\epsilon}{1-\epsilon} \cdot \left(\sum_{r=1}^K \frac{a_r'}{a_k'}\right)^{-1} \leq (1 + 4\epsilon) \cdot x_k'.$$

$$x_k = \left(\sum_{r=1}^K \frac{a_r}{a_k}\right)^{-1} \geq \left(\sum_{r=1}^K \frac{a_r'}{a_k'} \cdot \frac{1+\epsilon}{1-\epsilon}\right)^{-1} \geq \frac{1-\epsilon}{1+\epsilon} \cdot \left(\sum_{r=1}^K \frac{a_r'}{a_k'}\right)^{-1} \geq (1 - 4\epsilon) \cdot x_k'. \quad \square$$

Define $a_{tk}(\alpha) = \exp\left(-\sum_i \alpha_i \cdot \frac{v_{itk}}{\textbf{m}_t(v)}\right)$. Then, we have

$$\frac{a_{tk}(\alpha^*)}{a_{tk}(\tilde{\alpha})} = \exp\left(-\sum_i (\alpha_i - \tilde{\alpha}_i) \cdot \frac{v_{itk}}{\textbf{m}_t(v)}\right).$$

By the assumption that the instance is balanced, we have $\frac{v_{itk}}{\textbf{m}_t(v)} \leq \frac{d^2}{\epsilon^2}$, which implies

$$1 - 2\epsilon \leq \exp(-\epsilon) \leq \frac{a_{tk}(\alpha^*)}{a_{tk}(\tilde{\alpha})} \leq \exp(\epsilon) \leq 1 + 2\epsilon.$$

Thus, by Claim 4.9, the assignment fractions using $\tilde{\alpha}$ differ from those using $\alpha$ by at most $(1 \pm 4\epsilon)$. $\quad \square$

Finally, we now put all the pieces together to establish Lemma 4.4:

*Proof of Lemma 4.4.* Fix a balanced instance $I(v, K)$ and $\epsilon > 0$. By Lemma 4.6, there exists a parameter vector $\alpha^{(\epsilon)}$ such that $\alpha_i^{(\epsilon)} \in \left[0, \frac{d^2}{\epsilon} \cdot \ln\left(\frac{d}{\epsilon}\right)\right]$ for all $i \in [d]$. Therefore, there exists a vector $\tilde{\alpha} \in \textbf{NET}\left(\frac{d^2}{\epsilon} \cdot \ln\left(\frac{d}{\epsilon}\right), \frac{d^3}{\epsilon^3}\right)$ such that $|\tilde{\alpha}_i - \alpha_i^{(\epsilon)}| \leq \frac{\epsilon^3}{d^3}$ for all $i \in [d]$. By Lemma 4.8, the exponential assignment rule with $\tilde{\alpha}$ achieves a $(1 + O(\epsilon))$-approximation. $\quad \square$

# 5 Applications

Our framework applies broadly to a variety of online allocation problems. We briefly highlight two representative applications: online routing and Nash Social Welfare maximization.

**Online Routing.** In online routing (e.g., [3]), requests arrive over time and must be assigned to paths in a network to minimize congestion or delay. Each option (path) contributes load to

network edges, and the goal is to minimize the maximum edge utilization, which corresponds to the **MinMax** objective. Our approach uses a vector of learned edge-weight parameters to guide allocation decisions, combining strong worst-case guarantees with improved performance when input patterns are predictable.

Formally, we are given as input a directed graph $G = (V, E)$. At each time step $t \in [T]$, a flow request of amount $r_t$ between two vertices $\mathbf{s}_t$ and $\mathbf{t}_t$, along with a set of candidate paths $\mathcal{P}_t = \{P_{t,1}, P_{t,2}, \dots\}$, is revealed. Here, each $P_{t,k}$ is a path from $\mathbf{s}_t$ to $\mathbf{t}_t$ in the graph. The algorithm must assign values $q_{t,k}$ to each path such that $\sum_k q_{t,k} = r_t$. The load on an edge $e \in E$ is given by $\ell_e(q) = \sum_t \sum_{P_{t,k} | e \in P_{t,k}} q_{t,k}$, and the objective is to minimize the maximum congestion on any edge, i.e., $\min \max_{e \in E} \{\ell_e(q) \mid \sum_k q_{t,k} = r_t \text{ for all } t \in [T]\}$.

We note that online routing is a special case of the online allocation problem, where $A_t \subseteq \left\{ z_t \cdot \vec{f} \mid \vec{f} \in \{0,1\}^d, \ z_t \in \mathbb{R}_+ \right\}$, and the objective is to minimize $f(v) = \max_{i \in [d]} v_i$.

Our main result for online routing follows as a corollary of Theorem 4.1.

**Corollary 5.1.** *Given an instance of online routing and any $\epsilon > 0$, there exists a set of parameters $\alpha \in \mathbf{NET}\left( \frac{d^2}{\epsilon} \cdot \ln\left(\frac{d}{\epsilon}\right), \frac{d^3}{\epsilon^3} \right)$ such that the fractional solution $x_{tk} \propto \exp\left( -\sum_i \alpha_i \cdot f_{itk} \right), \quad \text{for } t \in [T]$, approximates the optimal objective within a factor of $(1 + O(\epsilon))$.*

**Online Nash Social Welfare.** As mentioned, our scheme also applies to maximation problems where, at each step, a divisible resource has to be distributed among a set of $d$ agents. The objective is to design an online algorithm that balances fairness and efficiency. At the start of each step $t$, the algorithm observes the value $v_{i,t}$ of each agent $i$ for that resource, and then irrevocably determines the allocation without knowledge of future values. If agent $i$ is allocated a fraction $x_{i,t}$, their utility increases by $x_{i,t} v_{i,t}$. The total utility of agent $i$ is then given by $u_i(\mathbf{x}) = \sum_t v_{i,t} x_{i,t}$.

The *Nash Social Welfare* (NSW) objective is known to provide a natural balance between fairness and efficiency. It is defined as the geometric mean of the agents' utilities: $\mathrm{NSW}(\mathbf{x}) = \left( \prod_i u_i(\mathbf{x}) \right)^{1/d}$. By applying our method for maximizing well-behaved objectives, we obtain the following corollary:

**Corollary 5.2.** *Given an instance of online NSW maximization and any $\epsilon > 0$, there exists a set of parameters $\alpha \in \mathbf{NET}\left( \frac{d^2}{\epsilon} \cdot \ln\left(\frac{d}{\epsilon}\right), \frac{d^3}{\epsilon^3} \right)$ such that the fractional solution $x_{t,k} \propto \exp\left( \sum_i \alpha_i \cdot v_{i,t,k} \right), \quad \text{for } t \in [T]$, approximates the optimal objective within a factor of $(1 - O(\epsilon))$.*

Note that using our results, we may further generalize this to options with vector utilities, i.e., where an option can add (possibly different amounts of) value to multiple agents simultaneously.

## 6 Closing Remarks

In this paper, we gave a general technique for designing online algorithms with predictions that applies to the entire spectrum of problems that can be modeled via the online covering framework. This has value in two respects. First, it gave the first learning-augmented results for important problems like online routing and scheduling that goes beyond simple assignment, and were beyond the scope of previous techniques. Second, and perhaps more importantly, it opens the door to the design of even more general-purpose methods for the design of online algorithms with predictions. For instance, can we give a technique for learning-augmented algorithms that applies to any covering LP where each online step reveals a new constraint in the LP? Techniques such as the online primal dual method that apply to this class of LPs have been hugely influential in the classical online algorithms literature (without predictions), which makes it a tempting proposition to explore similarly powerful tools in the learning-augmented setting.

## Acknowledgments and Disclosure of Funding

Ilan Reuven Cohen's research was supported by the Israel Science Foundation grant No. 1737/21. Debmalya Panigrahi's research was supported in part by NSF grants CCF-1955703 and CCF-2329230.

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

# A  Preprocessing for the MinMax Objective

We begin by introducing a preprocessing step for the **MinMax** objective. Given an instance $I(v, K)$, let **MinMax**$(v, K)$ denote the optimal value of the corresponding **MinMax** objective. Specifically, we construct an instance $I(\tilde{v}, \tilde{K})$ that $\epsilon$-approximates a given instance $I(v, K)$ as follows:

**Definition A.1.** An instance $I(\tilde{v}, \tilde{K})$ is said to $\epsilon$-approximate an instance $I(v, K)$ if:

- $\tilde{K}(j) \subseteq K(j)$ for all $j \in N$,

- **MinMax**$(\tilde{v}, \tilde{K}) \leq$ **MinMax**$(v, K) \cdot (1 + \epsilon)$,

- For any feasible allocation $\tilde{x}$ for $I(\tilde{v}, \tilde{K})$, if the load in any dimension is at most $L$ with respect to $\tilde{v}$, then the corresponding load with respect to $v$ is at most $L \cdot (1 + \epsilon)$.

(Note that $\tilde{x}$ remains a feasible allocation for $I(v, K)$ since $\tilde{K}(j) \subseteq K(j)$.)

Using this definition, we prove that in order to achieve a $(1 + O(\epsilon))$ to the instance $I(v, K)$ it is sufficent to acheive $(1 + \epsilon)$ approximation to $I(\tilde{v}, \tilde{K})$.

**Corollary A.2.** *Let $I(v, K)$ be a given instance. Suppose the instance is transformed into an $\epsilon$-approximate instance $I(\tilde{v}, \tilde{K})$. If a $(1 + \epsilon)$-approximate allocation is computed for the transformed instance $I(\tilde{v}, \tilde{K})$, then this allocation guarantees a $(1 + O(\epsilon))$-approximation for the original instance $I$.*

*Proof.* Let $x$ be a feasible solution for $I(\tilde{v}, \tilde{K})$ such that

$$\sum_t \sum_{k \in \tilde{K}(t)} x_{tk} \cdot \tilde{v}_{itk} \leq \mathbf{MinMax}(\tilde{v}, \tilde{K}) \cdot (1 + \epsilon).$$

Using Definition A.1, we obtain:

$$\sum_t \sum_{k \in \tilde{K}(t)} x_{tk} \cdot v_{itk} \leq \mathbf{MinMax}(\tilde{v}, \tilde{K}) \cdot (1 + \epsilon)^2$$

$$\leq \mathbf{MinMax}(v, K) \cdot (1 + \epsilon)^3.$$

$\square$

We now show that each of the three steps defined in the transformation produces an $\epsilon$-approximate instance with respect to the original instance.

**Claim A.3** (Step (1) of the preprocessing). *Given an instance $I(v, K)$ let $\hat{v}$ such that*

$$\hat{v}_{itk} = \begin{cases} 0 & \text{if } \frac{v_{itk}}{\max_{i'} v_{i'tk}} < \frac{\epsilon}{d}, \\ v_{itk} & \text{otherwise}. \end{cases}$$

*the the instance $I(\hat{v}, K)$ $\epsilon$-approximate the instance $I(v, K)$.*

*Proof.* By definition, **MinMax**$(\tilde{v}, \tilde{K}) \leq$ **MinMax**$(v, K)$. Next, given a solution $x$ such that for all $i \in [d]$,

$$\sum_{tk} x_{tk} \cdot \tilde{v}_{itk} \leq L,$$

we prove that

$$\sum_t \sum_{k \in \tilde{K}(t)} x_{tk} \cdot v_{itk} \leq L^* \cdot (1 + \epsilon).$$

For a fixed coordinate $i \in [d]$, define

$$R_i(t) = \{k \in K(t) : v_{itk} > \tilde{v}_{itk}\},$$

which represents the set of options for item $t$ where the load on coordinate $i$ is modified by the transformation. For an item $t$ and an option $k \in K(t)$, let

$$i_{tk}^{\max} \in \arg \max_{i' \in [d]} v_{i'tk}$$

denote the coordinate that experiences the maximum load for option $k$ of item $t$.

Observe that if $k \in R_i(t)$, then

$$v_{itk} \leq \frac{\epsilon}{d} \cdot v_{i_{tk}^{\max},t,k}.$$

Thus, we have:

$$\sum_{tk} x_{tk} \cdot v_{itk}$$

$$= \sum_{tk} x_{tk} \cdot \tilde{v}_{itk} + \sum_{t} \sum_{k \in R_i(t)} x_{tk} \cdot v_{itk}$$

$$\leq L + \sum_{t} \sum_{k \in R_i(t)} x_{tk} \cdot v_{itk}$$

$$= L + \sum_{i' \in d} \sum_{t} \sum_{k \in R_i(t) | i' = i_{tk}^{\max}} x_{tk} \cdot v_{itk}$$

$$\leq L + \sum_{i' \in d} \sum_{t} \sum_{k \in R_i(t) | i' = i_{tk}^{\max}} x_{tk} \cdot \frac{\epsilon}{d} \cdot \tilde{v}_{i'tk}$$

$$= L + \sum_{i' \in d} \sum_{t} \sum_{k \in R_i(t) | i' = i_{tk}^{\max}} \frac{\epsilon}{d} \cdot x_{tk} \cdot \tilde{v}_{i'tk}$$

$$\leq L + \frac{\epsilon}{d} \cdot \sum_{i' \in d} \sum_{tk} x_{tk} \cdot \tilde{v}_{i'tk}$$

$$\leq L + \sum_{i' \in [d]} L \cdot \frac{\epsilon}{d}$$

$$= L \cdot (1 + \epsilon),$$

where the first and final inequalities follow from the definition of $x$, and the second inequality follows from the transformation definition. □

The

**Claim A.4** (Step (2) of the Preprocessing). *Given an instance $I(v, K)$, let $I(\tilde{v}, K')$ be the instance obtained by removing option $k'$ from item $t$ whenever there exists another option $k \in K(t)$ such that*

$$v_{t,k'}^{\max} > \frac{d}{\epsilon} \cdot v_{t,k}^{\max},$$

*where $v_{t,k}^{\max} = \max_{i \in [d]} v_{i,t,k}$.*

*Then $I(v, K')$ is an $\epsilon$-approximation of the instance $I(v, K)$.*

*Proof.* Clearly, an assignment with a load of at most $L$ for all $i \in [d]$ in $I(\tilde{v}, \tilde{K})$ results in a load of at most $L$ in $I(v, K)$, since the retained options have identical loads. We now show that if an allocation $x$ achieves a makespan of $L$ in $I(v, K)$, then there exists a transformed allocation $\hat{x}$ that achieves a makespan of at most $L \cdot (1 + \epsilon)$ in $I(\tilde{v}, \tilde{K})$.

We explicitly construct such an assignment $\hat{x}$ and show that it satisfies the required bound. Define

$$k_t^{\min} \in \arg \min_k v_{t,k}^{\max}$$

as the option with the smallest maximum load for item $t$, breaking ties by selecting the smallest index. Let

$$i_{tk}^{\max} = \arg\max_i v_{i,t,k}$$

denote the coordinate where option $k$ of item $t$ imposes the maximum load.

Next, define the set

$$R(t) = \{k \in K(t) \mid v_{t,k}^{\max} > \frac{d}{\epsilon} \cdot v_{t,k_t^{\min}}^{\max}\},$$

which identifies options with disproportionately high loads relative to the smallest maximum load. We then construct the transformed allocation $\hat{x}$ for each item $t \in [n]$ as follows:

$$\hat{x}_{tk} = \begin{cases} x_{tk} + \sum_{k' \in R(t)} x_{tk'} & \text{if } k = k_t^{\min}, \\ 0 & \text{if } k \in R(t), \\ x_{tk} & \text{otherwise.} \end{cases}$$

This transformation ensures that options in $R(t)$ are reallocated to the option with the smallest maximum load, maintaining feasibility while ensuring that the makespan increases by at most a factor of $(1 + \epsilon)$. Specifically, we have:

$$
\begin{aligned}
&\sum_{tk} \hat{x}_{tk} \cdot v_{itk} \\
&= \sum_{tk} x_{tk} \cdot v_{itk} + \sum_{t} \sum_{k' \in R(t)} x_{tk'} \cdot v_{i,t,k_t^{\min}} \\
&\leq L + \sum_{t} \sum_{k' \in R(t)} x_{tk'} \cdot \frac{\epsilon}{d} \cdot v_{i_{tk'}^{\max},t,k'} \\
&= L + \frac{\epsilon}{d} \cdot \sum_{i',t} \sum_{k' \in R(t)} \sum_{i'=i_{tk'}^{\max}} x_{tk'} \cdot v_{i',t,k'} \\
&\leq L + \frac{\epsilon}{d} \cdot \sum_{i'} \sum_{t,k} x_{tk} \cdot v_{i',t,k} \\
&\leq L + \sum_{i' \in [d]} L \cdot \frac{\epsilon}{d} \\
&= L \cdot (1 + \epsilon),
\end{aligned}
$$

where the first and final inequalities follow from the definition of $x$, and the second inequality follows from the fact that for $k' \in R(t)$,

$$v_{i,t,k_t^{\min}} \leq v_{t,k_t^{\min}}^{\max} < \frac{\epsilon}{d} \cdot v_{t,k'}^{\max} = \frac{\epsilon}{d} \cdot v_{i_{tk'}^{\max},t,k'}.$$

$\square$

**Claim A.5** (Step (3) of the Preprocessing). *Given an instance $I(v, K)$*

*if $v_{itk} = 0$, then set $\tilde{v}_{itk} = 0$; otherwise, set*

$$\tilde{v}_{itk} = \mathbf{m}_t(v) \cdot (1 + \epsilon)^{\left\lfloor \log_{1+\epsilon}\left( \frac{\hat{v}_{itk}}{\mathbf{m}_t(v)} \right) \right\rfloor}.$$

*Then $I(\tilde{v}, K)$ is an $\epsilon$-approximation of the instance $I(v, K)$.*

*Proof.* By definition,

$$\tilde{v}_{itk} \leq v_{itk} \leq (1 + \epsilon) \cdot \tilde{v}_{itk}.$$

Therefore, $I(\tilde{v}, K)$ is an $\epsilon$-approximation of the instance $I(v, K)$.

$\square$

By applying Claim A.3,Claim A.4,Claim A.5, and Corollary A.2 we proved the Lemma 4.3.

# B  Perturbation and Sensitivity Analysis

One of our main tools for bounding learned parameters is perturbation and sensitivity analysis, following the framework of Boyd et al. [16].

## B.1  Perturbed Convex Programs

Consider a convex program represented by a tuple $(f, h)$, where $f_i : \mathbb{R}^d \to \mathbb{R}$ for $i \in [0, \tilde{m}]$ and $h_i : \mathbb{R}^d \to \mathbb{R}$ for $i \in [1, \tilde{n}]$, formulated as:

$$\begin{aligned}
&\text{minimize } f_0(x) \\
&\text{subject to } f_i(x) \le 0, \quad i \in [1, \tilde{m}] \\
&\qquad\qquad\; h_i(x) = 0, \quad i \in [1, \tilde{n}].
\end{aligned}$$

We define the perturbed version of this problem as follows. Let $u \in \mathbb{R}^{\tilde{m}}$ and $v \in \mathbb{R}^{\tilde{n}}$. Define perturbed constraints:

$$f_i'(x) = f_i(x) - u_i, \quad h_i'(x) = h_i(x) - v_i.$$

The perturbed problem becomes:

$$\begin{aligned}
&\text{minimize } f_0(x) \\
&\text{subject to } f_i(x) \le u_i, \quad i \in [1, \tilde{m}] \\
&\qquad\qquad\; h_i(x) = v_i, \quad i \in [1, \tilde{n}].
\end{aligned}$$

This coincides with the original problem when $u = 0$ and $v = 0$. Positive $u_i$ relaxes the $i$th inequality; negative $u_i$ tightens it. The vector $v$ perturbs the right-hand sides of the equality constraints.

Let $p^*(u, v)$ denote the optimal value of the perturbed problem. If the problem is infeasible, we define $p^*(u, v) = \infty$. When the original problem is convex, $p^*(u, v)$ is convex in both $u$ and $v$.

## B.2  A Global Inequality via Duality

Assume strong duality holds and that the dual optimum is attained (which is the case under Slater's condition). Let $(\lambda^*, \nu^*)$ denote an optimal dual solution to the original problem. Then the following global bound holds.

**Lemma B.1** (Perturbation Inequality, [16]). *For all $u \in \mathbb{R}^{\tilde{m}}$ and $v \in \mathbb{R}^{\tilde{n}}$,*

$$p^*(0, 0) \ge p^*(u, v) - \lambda^{*T} u - \nu^{*T} v.$$

## B.3  Proof Sketch

To derive this inequality, consider any feasible solution $x$ to the perturbed problem, i.e.,

$$f_i(x) \le u_i, \quad \forall i \in [1, \tilde{m}], \qquad h_i(x) = v_i, \quad \forall i \in [1, \tilde{n}].$$

By strong duality:

$$\begin{aligned}
p^*(0, 0) &= g(\lambda^*, \nu^*) \\
&\le f_0(x) + \sum_{i=1}^{\tilde{m}} \lambda_i^* f_i(x) + \sum_{i=1}^{\tilde{n}} \nu_i^* h_i(x) \\
&\le f_0(x) + \lambda^{*T} u + \nu^{*T} v,
\end{aligned}$$

where the last inequality follows from $f_i(x) \le u_i$, $h_i(x) = v_i$, and $\lambda^* \ge 0$.

Rearranging, we conclude:

$$f_0(x) \ge p^*(0, 0) - \lambda^{*T} u - \nu^{*T} v.$$

This inequality quantifies how much the objective value may change in response to perturbations of the constraints, depending linearly on the dual multipliers.

## C  Learnability of the Prediction Vectors

We consider the learning model introduced by [24], and show that under this model, the dual vector $\alpha$ can be learned efficiently from sampled instances. Specifically, we consider the following model: the $t$th step (i.e., the values of $v_t = (v_{i,t,k} : i \in [d], k \in K(t))$ is independently sampled from a (discrete) distribution $\mathcal{D}_j$.

We set up the online allocation problem for the **MinMax** objective; the setup for the **MaxMin** objective is very similar and is omitted for brevity. Let $L = \mathbb{E}_{P \sim \mathcal{D}}[\textbf{MinMax}(P)]$ be the expected value of the **MinMax** objective in the optimal solution for an instance $P$ drawn from $\mathcal{D}$.

Morally, we would like to say that we can obtain the vector $\alpha$ that gives a nearly optimal solution (in expectation) using vector allocation (i.e., a **MinMax** objective of $(1 + \epsilon) \cdot L$ in expectation for some error parameter $\epsilon$) using a bounded (as a function of $\epsilon$) number of samples. Similar to [24], we need the following assumption:

**Small Items Assumption:** Conceptually, this assumption states that each individual item has a small utility compared to the overall utility of any agent in an optimal solution. Precisely, we need $v_{itk} \leq \frac{L}{\zeta}$ for every $i \in [d], t \in [T]$, and $k \in K(t)$ for some value $\zeta = \Theta\left(\frac{\log d}{\epsilon^2}\right)$.

We show the following PAC learning theorem for the **MinMax** objective:

**Theorem C.1.** *Fix an $\epsilon > 0$ for which the small items assumption holds. Then, there is an (learning) algorithm that samples $O(\frac{d}{\log d} \cdot \log \frac{d}{\epsilon})$ independent instances from $\mathcal{D}$ and outputs (with high probability) a prediction vector $\alpha$ such that using $\alpha$ in the allocation scheme gives a **MinMax** objective of at least $(1 + O(\epsilon)) \cdot L$ in expectation over instances $P \sim \mathcal{D}$.*

*Proof Sketch.* Recall that in PAC theory, the number of samples needed to learn a function from a family of $N$ functions is about $O(\log N)$. Indeed, restricting $\alpha$ to be in the class **NET**$(K, S)$ serves this role of limiting the hypothesis class to a finite, bounded set since $|\textbf{NET}(K, S)| = (K \cdot S)^d$ where $S = K = O(\text{poly}(d, \epsilon))$. Using standard PAC theory, this implies that using about $O(d \log K) = O(d \cdot \log \frac{d}{\epsilon})$ samples, we can learn the "best" vector in **NET**$(K, S)$. Our main technical work is to show that this "best" vector produces an approximately optimal solution when used.

$\square$

### C.1  Details of PAC Learning

Given an instance $P$, Let $\ell_i(P, \alpha)$ the value of the $i$th dimension of $\mathbf{v}_{\text{tot}}^f$ after applying our scheme on the instance $P$ with the parameters $\alpha$. Let $K = \frac{d^2}{\epsilon}$ and $S = \frac{d^3}{\epsilon^3}$.

Let us consider a combination of all instances in the support of the distribution $\mathcal{D}$. For $L$ processing matrices $P^{(1)}, P^{(2)}, \ldots, P^{(L)}$. We define $P^{\text{all}} = \bigoplus_{r=1}^{L} P^{(r)}$ to be the instance defined by the $n \cdot L$ items. For every $\ell \in [L]$ and $t \in [T]$, we have a step $t^{(\ell)}$ with values $v_t^{(\ell)}$.

The following observation is immediate (subadditivity):

**Observation C.2.** $\textbf{MinMax}(P^{all}) \leq \sum_{r=1}^{L} \textbf{MinMax}(P^{(r)})$.

Using this observation, we can prove the following lemma, by considering the combination of all instances in $\mathcal{D}$, scaled by their respective probabilities.

**Lemma C.3.** *There exists $\alpha \in \textbf{NET}(K, S)$, such that for every $i \in [d]$, we have*

$$\mathbb{E}_{P \sim \mathcal{D}}[\ell_i(\alpha)] \leq (1 + \epsilon) \cdot L^*.$$

*Proof.* Consider the instance $\mathbb{P} = \bigoplus \Pr_D[P] \cdot P$ where $\Pr_D[P]$ is the probability mass of $P$ in $\mathcal{D}$, and $\Pr_D[P] \cdot P$ is the matrix $P$ multiplied by $\Pr_D[P]$. By Observation C.2, we have

$$\textbf{MinMax}(\mathbb{P}) \leq \sum_P \Pr_{\mathcal{D}}[P]\textbf{MinMax}(P)$$

$$= \mathbb{E}_{P \sim \mathcal{D}}[\textbf{MinMax}(P)] = L.$$

We can apply Lemma 4.4 to the combined instance to show there exists $\alpha^* \in \mathbf{NET}(K, S)$ such that for every $i \in [d]$, we have,

$$\sum_{tk} x_{t,k}(\mathbb{P}, \alpha^*) \cdot p_{itk} \leq (1 + \epsilon)\mathbf{MinMax}(\mathbb{P}) \leq (1 + \epsilon) \cdot L^*$$

where $t$ indexes over all steps in $\mathbb{P}$. Notice that $x_{t,k}(\mathbb{P}, \alpha^*)$ depends on the utility vector for step $t$, which is part of the instance $P \in \mathcal{D}$ that $t$ belongs to. Therefore, the left side of the above inequality is exactly

$$\sum_P \sum_{t,k} x_{t,k}(P, \alpha^*) \cdot \mathrm{Pr}_\mathcal{D}[P] \cdot p_{i,t,k}$$

$$= \mathbb{E}_{P \sim \mathcal{D}} \sum_{t,k} x_{t,k}(P, \alpha^*) p_{i,t,k}$$

$$= \mathbb{E}_{P \sim \mathcal{D}} \sum_{t \in [T]} \ell_i(P, \alpha^*),$$

as required.

$\square$

For any real numbers $A, B, \epsilon, C$, we use $A \approx_{\epsilon,C} B$ to denote $|A - B| \leq \epsilon \cdot \max(B, C)$. The next lemma appears in [26]:

**Lemma C.4** (Lemma D.6 in in [26]). *For any $\alpha \in \mathbf{NET}(K, S)$, with high probability over $P \sim \mathcal{D}$, we have*

$$\forall i \in [d] : \ell_i(P, \alpha) \approx_{\epsilon, L^*} \mathbb{E}_{P' \sim D} \ell_i(P', \alpha).$$

**The learning algorithm.** We sample $H = O\left(\frac{d}{\log d} \log \frac{d}{\epsilon}\right)$ instances $P_1, P_2, \ldots, P_H$ independently and randomly form $\mathcal{D}$. We output $\tilde{\alpha} \in \mathbf{NET}(K, S)$ that maximizes $\min_{i \in [d]} \frac{1}{H} \sum_{h=1}^H \ell_i(P_h, \tilde{\alpha})$.

The next lemma also appears in [26]:

**Lemma C.5** (Lemma D.7 in [26]). *With probability at least $1 - \frac{1}{(K \cdot S)^m}$, for every $\alpha \in \mathbf{NET}(K, S)$ and for every $i \in [d]$, we have*

$$\frac{1}{H} \sum_{h=1}^H \ell_i(P_h, \alpha) \approx_{\epsilon, L^*} \mathbb{E}_{P \sim D} \ell_i(P, \alpha).$$

Now assume the event in Lemma C.5 happens. Then by Lemma C.3, there exists some $\alpha \in \mathbf{NET}(K, S)$ such that

$$\min_{i \in [m]} \frac{1}{H} \sum_{h=1}^H \ell_i(P_h, \alpha) \leq (1 + \epsilon)^2 \cdot L^*.$$

In particular, since $\tilde{\alpha}$ maximizes $\min_{i \in [m]} \frac{1}{H} \sum_{h=1}^H \ell_i(P_h, \tilde{\alpha})$ for $\tilde{\alpha} \in \mathbf{NET}(m, \epsilon)$, we can conclude that

$$\min_{i \in [m]} \frac{1}{H} \sum_{h=1}^H \ell_i(P_h, \tilde{\alpha}) \leq (1 + \epsilon)^2 \cdot L^*.$$

Applying Lemma C.5 again, we get

$$\min_{i \in [m]} \mathbb{E}_{P \sim \mathcal{D}} \ell_i(P, \tilde{\alpha}) \leq (1 + \epsilon)^3 \cdot L^*.$$

We now apply Lemma C.4 to $\tilde{\alpha}$. We have that with high probability over $P \sim \mathcal{D}$, for every $i \in [m]$ the following holds:

$$\ell_i(P, \tilde{\alpha}) \leq$$
$$\mathbb{E}_{P' \sim \mathcal{D}} \ell_i(P', \tilde{\alpha}) + \epsilon \cdot \max\{L^*, \mathbb{E}_{P' \sim \mathcal{D}} \ell_i(P', \tilde{\alpha})\} \leq$$
$$(1 + \epsilon)^4 \cdot L^*.$$

Therefore, $\mathbf{MinMax}(P, \tilde{\alpha}) \leq (1 + \Omega(\epsilon)) \cdot L^*$. This completes the proof of Theorem C.1.

# D  Robustness–Consistency Tradeoff

In this section, we show that our learning-augmented scheme can be modified to balance consistency and robustness, achieving near-optimal performance when predictions are accurate while retaining strong worst-case guarantees when they are not.

Recall that an algorithm is said to be $\gamma$-*consistent* and $\delta$-*robust* if it achieves a $\gamma$-approximation under accurate predictions (consistency), and a $\delta$-approximation in the worst case when predictions are unreliable (robustness).

For allocation problems with minimization objectives, the worst-case approximation ratio without any predictions is $O(\log d)$, and for maximization objectives it is $O(d)$, where $d$ is the number of agents.

We show that our learning-augmented scheme can be modified to satisfy this robustness–consistency tradeoff.

**Modified Algorithm for Minimization Objectives.**  Let $\alpha \in \mathbb{R}^d$ be the predicted parameter vector. The algorithm operates in two phases:

1. **Prediction Phase:** At each time step, use the exponential assignment scheme with parameters $\alpha$.
2. **Fallback Phase:** Monitor the cumulative objective value. If it exceeds the optimal value by a factor larger than $O(\log d)$, the algorithm switches to a standard worst-case online algorithm.

Let $\eta$ be the approximation factor achieved using $\alpha$. Then, the final approximation ratio is $\min(\eta, O(\log d))$, ensuring both consistency and robustness.

**Modified Algorithm for Maximization Objectives.**  Let $\alpha \in \mathbb{R}^d$ be the predicted parameter vector, and let $\lambda \in [0, 1]$ be a confidence parameter reflecting trust in the prediction. The algorithm allocates each item as follows:

1. Allocate a fraction $1 - \lambda$ of the item using the exponential assignment scheme with parameters $\alpha$.
2. Allocate the remaining $\lambda$ fraction using a worst-case robust algorithm (e.g., uniform allocation or greedy).

This strategy guarantees:

- **Consistency:** The portion allocated by the learned parameters achieves an approximation ratio of $(1 - \lambda)(1 - \epsilon)$, assuming the predicted parameters yield a $(1 - \epsilon)$-approximation.
- **Robustness:** The worst-case portion contributes at most $\lambda \cdot d$, matching the lower bound of known worst-case algorithms.

Hence, the algorithm achieves a $(1 - \lambda)(1 - \epsilon)$-consistent and $\lambda \cdot d$-robust guarantee.

# E  Learning-Augmented Online Allocation for the MaxMin Objective

In this section, we prove our main result for the **MaxMin** objective.

**Theorem E.1.** *Given an instance of the online allocation problem with a **MaxMin** objective and any $\epsilon > 0$, there exists a set of learned parameters*

$$\alpha \in \textbf{NET}\left( \frac{d^2}{\epsilon} \cdot \log\left(\frac{d}{\epsilon}\right), \ \frac{d^3}{\epsilon^3} \right)$$

*and an online algorithm that uses the exponential assignment scheme with $-\alpha$, such that the resulting fractional solution is a $(1 - O(\epsilon))$-approximation.*

To prove this theorem, we proceed similarly to the **MinMax** case: we begin with a preprocessing step that transforms the instance into a *balanced* form. However, the details of the preprocessing differ in the **MaxMin** setting.

### E.1 Preprocessing for the MaxMin Objective

We begin by describing a transformation that modifies the input instance into a balanced form suitable for our learning-augmented algorithm. The transformation has two steps:

**Step 1: Remove agents with large monopolist values.** For each step $t \in [T]$, define

$$v^*_{it} = \max_{k \in K(t)} v_{itk}, \quad k^*_{it} = \arg\max_{k \in K(t)} v_{itk}.$$

Define the *monopolist value* of agent $i$ as $a_i = \sum_{t \in [T]} v^*_{it}$, and let $a_{\min} = \min_{i \in [d]} a_i$. For each agent $i$ such that

$$a_i \geq \frac{d}{\epsilon} \cdot a_{\min},$$

we allocate an $\epsilon/d$ fraction of the resource at each step to their preferred option $k^*_{it}$.

**Step 2: Zero out negligible values.** For the remaining instance (with the reduced set of agents), define a new instance $\hat{v}$ by zeroing out small entries:

$$\hat{v}_{itk} = \begin{cases} 0 & \text{if } \frac{v_{itk}}{\max_{i'} v_{i'tk}} < \frac{\epsilon}{d}, \\ v_{itk} & \text{otherwise.} \end{cases}$$

Let $J_i(t) \subset K(t)$ be the set of options that were modified for agent $i$ at step $t$, and define

$$\tilde{v}_{it} = \max_{k \in J_i(t)} v_{itk}.$$

**Step 3: Quantize values to form a balanced instance.** Define the final preprocessed values $\tilde{v}$ as:

$$\tilde{v}_{itk} = \begin{cases} 0 & \text{if } \hat{v}_{itk} = 0, \\ \mathbf{m}_t(v) \cdot (1 + \epsilon)^{\left\lfloor \log_{1+\epsilon}\left(\frac{v_{itk}}{\mathbf{m}_t(\tilde{v})}\right) \right\rfloor} & \text{otherwise.} \end{cases}$$

This rounding ensures that all nonzero values are restricted to a logarithmic grid defined by the base $1 + \epsilon$, thereby yielding a *balanced* instance.

**Lemma E.2.** *Let $I(v, K)$ be an instance of the allocation problem with the **MaxMin** objective, and let $\epsilon > 0$. Then, the transformed instance $I(\tilde{v}, K)$, obtained via the three-step preprocessing, satisfies the following: any $(1 - \epsilon)$-approximate solution for $I(\tilde{v}, K)$, when combined with the allocations reserved in Step 1, yields a $(1 - O(\epsilon))$-approximate solution for the original instance $I(v, K)$.*

*Proof.* We begin by analyzing the impact of Step 1. The utility of any agent $i$ removed during this step is guaranteed to be at least

$$\frac{\epsilon}{d} \cdot a_i \geq a_{\min} \geq \mathbf{MaxMin}(v, K),$$

so these agents are fully satisfied by the reserved allocation. Moreover, the total amount of resource allocated to these agents is at most an $\epsilon$-fraction of the total, ensuring that the remaining instance is affected by at most a $(1 - \epsilon)$ loss in the objective value.

Now consider the remaining agents in the modified instance $I(\hat{v}, K)$. In Step 2, we zero out negligible values to reduce the dynamic range. Let $\tilde{v}_{it} = \max_{k \in J_i(t)} v_{itk}$, where $J_i(t)$ is the set of coordinates zeroed out for agent $i$ at step $t$. For each $i \in [d]$, we bound the total value removed as:

$$\sum_t \tilde{v}_{it} \leq \frac{\sum_t \max_{i',k'} v_{i'tk'}}{d/\epsilon} \leq \frac{\sum_{i'} a_{i'}}{d/\epsilon} \leq \epsilon \cdot \frac{a_{\min}}{d} \leq \epsilon \cdot \mathbf{MaxMin}(v, K),$$

where the last inequality uses the fact that $\mathbf{MaxMin}(v, K) \geq a_{\min}/d$, as each agent can receive a $1/d$ share of their monopolist option.

Hence, the approximation loss from zeroing out small values is bounded by $\epsilon \cdot \mathbf{MaxMin}(v, K)$, and the resulting instance $I(\hat{v}, K)$ differs from the original by at most an $O(\epsilon)$ factor.

Finally, Step 3 introduces a geometric rounding of values to the nearest power of $(1 + \epsilon)$. As shown in the **MinMax** case, this quantization step results in an additional loss of at most a $(1 - \epsilon)$ factor.

Combining the effects of the three steps, the overall degradation in objective value is at most a $(1 - O(\epsilon))$ factor. Thus, a $(1 - \epsilon)$-approximate solution to the preprocessed instance yields a $(1 - O(\epsilon))$-approximate solution for the original instance. $\qquad\square$

## E.2 Existence of Discretized Parameters for Balanced Instances for MaxMin

In light of Lemma E.2, to prove Theorem E.1. it suffices to consider balanced instances, as stated in the following lemma:

**Lemma E.3.** *Given a balanced instance of the online allocation problem with a **MaxMin** objective and $\epsilon > 0$, there exists a set of parameters*

$$\alpha \in \mathbf{NET}\left(\frac{d^2}{\epsilon} \cdot \ln\left(\frac{d}{\epsilon}\right), \frac{d^3}{\epsilon^3}\right)$$

*such that the fractional solution defined by the exponential assignment scheme with parameters $-\alpha$ is a $(1 - O(\epsilon))$-approximation to the optimal objective.*

As in the **MinMax** objective, we define a slightly perturbed convex program (see Figure 2). We use $L^*$ to denote the optimal **MinMax** value, and define the following convex program for $\epsilon > 0$:

$$\min \quad \sum_{t \in [T]} \mathbf{m}_t(v) \sum_{k \in K(t)} x_{tk} \ln\left(\frac{x_{tk}}{e}\right)$$

$$\text{s.t.} \quad \sum_{t \in [T]} \sum_{k \in K(t)} v_{itk} \cdot x_{tk} \geq L^* \cdot (1 - \epsilon), \qquad \forall i \in [d],$$

$$\sum_{k \in K(t)} x_{tk} = 1, \qquad \forall t \in [T],$$

$$x_{tk} \geq 0, \qquad \forall k \in K(t), t \in [T]$$

Figure 2: Convex Programming Formulation for the **MaxMin** Objective

**Lemma E.4.** *Given an instance of the online allocation problem with the **MaxMin** objective and any $\epsilon \geq 0$, there exists a vector $\alpha^{(\epsilon)} \in \mathbb{R}_+^d$ such that the fractional solution defined by the exponential assignment scheme with parameters $-\alpha^{(\epsilon)}$ is $(1 - O(\epsilon))$-approximately optimal.*

*Proof.* Given such instance and for fixed $\epsilon$ consider the convex program of Figure 2. By our assumption, $L^*$ is the optimal **MaxMin** objective therefore there exists a feasible solution for the convex program for any $\epsilon \geq 0$. Accordingly, define the Lagrangian $L(x, \alpha, \beta)$ as

$$\sum_{t \in [T]} \mathbf{m}_t(v) \sum_{k \in K(t)} x_{tk} \ln\left(\frac{x_{tk}}{e}\right) + \sum_{i \in [d]} \alpha_i \left(L^*(1-\epsilon) - \sum_{t \in [T]} \sum_{k \in K(t)} v_{itk} \cdot x_{tk}\right) + \sum_{t \in [T]} \beta_t \left(1 - \sum_{k \in K(t)} x_{tk}\right).$$

From the KKT conditions for the optimal solution to the convex program as a function of $\epsilon$ $x^{(\epsilon)}, \alpha^{(\epsilon)}, \beta^{(\epsilon)}$, the solution that allocates according to $x^{(\epsilon)}$ is a $(1 + \epsilon)$-approximation to the optimal objective $L^*$, and $\alpha_i^{(\epsilon)} \geq 0$ for all $i \in [d]$. Furthermore,

$$\frac{dL}{dx_{tk}} = 0 \quad \text{for all } k \in K(t), \text{ which gives } \mathbf{m}_t(v) \cdot \ln(x_{tk}^{(\epsilon)}) - \mathbf{m}_t(v) \cdot \sum_i \alpha_i^{(\epsilon)} \cdot v_{itk} = \beta_t^{(\epsilon)}.$$

For any two options $k, r \in K(t)$, we obtain:

$$\mathbf{m}_t(v) \cdot \ln(x_{tk}^{(\epsilon)}) - \sum_i \alpha_i^{(\epsilon)} \cdot v_{itk} = \mathbf{m}_t(v) \cdot \ln(x_{tr}^{(\epsilon)}) - \sum_i \alpha_i^{(\epsilon)} \cdot v_{itr}.$$

Therefore, $\ln\left(\dfrac{x_{tk}^{(\epsilon)}}{x_{tr}^{(\epsilon)}}\right) = \sum_i \alpha_i^{(\epsilon)} \cdot \dfrac{v_{itk}}{\mathbf{m}_t(v)} - \sum_i \alpha_i^{(\epsilon)} \cdot \dfrac{v_{itr}}{\mathbf{m}_t(v)}$. Coupled with $\displaystyle\sum_{k \in K(t)} x_{tk}^{(\epsilon)} = 1$, we get

$$x_{tk}^{(\epsilon)} \propto \exp\left(\sum_i \alpha_i^{(\epsilon)} \cdot \frac{v_{itk}}{\mathbf{m}_t(v)}\right). \qquad \square$$

**Bounding the Learned Parameters.** As in the **MinMax** objective, we bound the learned parameters using perturbation and sensitivity analysis techniques.

**Lemma E.5.** *Let $x^{(\epsilon)}, \alpha^{(\epsilon)}, \beta^{(\epsilon)}$ be the optimal solution to the convex program in Figure 1, for some $\epsilon > 0$. Then, for all $i \in [d]$, it holds that $\alpha_i^{(\epsilon)} \le \frac{d^2}{\epsilon} \cdot \log\left(\frac{d}{\epsilon}\right)$.*

*Proof.* We define a perturbed convex program based on Figure 2, where $u_i$ corresponds to the constraint $\alpha_i$.

For each $i \in [d]$, setting $u_i = -\epsilon \cdot L^*$ and $u_{i'} = 0$ for $i' \ne i$ ensures that constraint $i$ in the perturbed problem matches the original constraint, thereby guaranteeing a feasible solution. By Lemma B.1,

$$p^*(0,0) \ge p^*(u,v) + \alpha_i^{(\epsilon)} \cdot \epsilon \cdot L^*, \text{ which implies}$$

$$\alpha_i^{(\epsilon)} \cdot \epsilon \cdot L^* \le p^*(0,0) - p^*(u,v) \le \sum_{t \in [T]} \mathbf{m}_t(v) \ln |K(t)| \le d \cdot L^* \cdot \log\left(\frac{d}{\epsilon}\right),$$

where the second inequality follows from Claim 4.7, and the third inequality by $\sum_t \mathbf{m}_t(v) \le d \cdot L^*$ and $\log |K(t)| \le \log\left(\frac{d}{\epsilon}\right)$. $\qquad\square$

Finally, we now put all the pieces together to establish Lemma E.3:

*Proof of Lemma E.3.* Fix a balanced instance $I(v, K)$ and $\epsilon > 0$. By Lemma E.5, there exists a parameter vector $\alpha^{(\epsilon)}$ such that $\alpha_i^{(\epsilon)} \in \left[0, \frac{d^2}{\epsilon} \cdot \log\left(\frac{d}{\epsilon}\right)\right]$ for all $i \in [d]$. Therefore, there exists a vector $\tilde\alpha \in \mathbf{NET}\left(\frac{d^2}{\epsilon} \cdot \log\left(\frac{d}{\epsilon}\right), \frac{d^3}{\epsilon^3}\right)$ such that $|\tilde\alpha_i - \alpha_i^{(\epsilon)}| \le \frac{\epsilon^3}{d^3}$ for all $i \in [d]$. By Lemma 4.8, the exponential assignment rule with $\tilde\alpha$ achieves a $(1 + O(\epsilon))$-approximation. $\qquad\square$

## F   Generalization to Well-Behaved function

We now complete the proof of Theorem 4.1

*Proof of Theorem 4.1.* Fix an objective function $f$ and an instance $I(v, K)$. Let $\ell_i^f$ denote the load in the $i$th dimension in an optimal solution for objective function $f$. Also, let $x_{i,t}$ denote the fraction at step $t$ assigned to option $k$ in this optimal solution.

Now, consider the instance $\tilde{I}(\tilde{v}, K)$, where $\tilde{v}_{itk} = \frac{v_{itk}}{\ell_i^f}$. By the monotonicity property of $f$, the optimal objective value for $\tilde{I}$ is 1. Therefore, by Lemma 4.4, there exists $\tilde\alpha$ such that using an allocation, we get $\ell^*(\tilde{I}, \alpha) \ge 1 - \epsilon$ for maximization and $\ell^*(\tilde{I}, \alpha) \le 1 + \epsilon$ for minimization.

By the definition of the allocation, $x_{t,k}^*$ is proportional to

$$\exp\left(-\sum_i \tilde{v}_{i,t,k} \cdot \tilde\alpha_i\right) = \exp\left(-\sum_i v_{i,t,k} \cdot \frac{\tilde\alpha_i}{\ell_i^f}\right).$$

Thus, if we define $\alpha$ such that $\alpha_i = \frac{\tilde\alpha_i}{\ell_i^f}$, then the corresponding allocation gives a $(1 - \epsilon)$-approximate solution for maximization and a $(1 + \epsilon)$-approximate solution for minimization. $\qquad\square$

