# OpenReview forum: "A Learning-Augmented Approach to Online Allocation Problems"
_NeurIPS.cc/2025/Conference — NeurIPS 2025 poster_

### Official Review · Reviewer_NQoi · 2025-07-02

**Clarity:** 2
**Significance:** 2
**Originality:** 3
**Rating:** 3
**Confidence:** 4

**Summary:**

The paper considers the online resource allocation problem, where the objective is to maximize the cumulative reward of an allocation strategy, with each allocation yielding a corresponding reward. The focus is on learning-augmented algorithms, where machine learning is used to predict future inputs, and these predictions are incorporated into the algorithmic design. The paper analyzes the problem within a unified framework by considering the existence of learnable parameters and strategies that can achieve a $1 + O(\epsilon)$ approximation. It also examines the sensitivity and robustness of the algorithms under small perturbations to the input or predictions. The paper highlights two specific applications—online routing and Nash social welfare—and presents corollaries of the main theoretical results in these contexts.

**Questions:**

Please see the above weakness section.

**Ethical Concerns:**

["NO or VERY MINOR ethics concerns only"]

**Final Justification:**

I believe that the writing of the paper could benefit from substantial changes/rewriting, as the current version is not easy to follow and the necessary modifications remain unclear based on the current rebuttal. So I would recommend borderline reject, given that we would like the publications to be clear enough to the community, and to the audience.

**Quality:**

2

**Strengths And Weaknesses:**

Strength

- The technical results are new, especially the construction of the balanced instance,

- The paper considers a relatively general class of online resource allocation problem, and shows a unified framework of learning-augmented online algorithms, by characterizing the approximation ratio.

- The paper also conducts the sensitivity analysis by considering the perturbation.

Weakness

- The paper could benefit from another round of polishing and it current form at is less ready for publication, since the current structure is a bit hard to follow with respect to the logic connections. For example, the introduction part focuses on heavily on the applications, but the applications mentioned in Section 5 do not match these applications exactly. One can get easily lost in the last two paragraphs before the Organization in Introduction. These two paragraphs do not help introducing this paper and its main theme. Also, merging the problem formulation with the contributions is somewhat confusing. Having a separate section of the problem formulation would make things clearer.

- The proof of Lemma 4.4 is missing, or I may miss something. It would be helpful if the authors could point to the place.

- MaxMin and MinMax are not defined with mathematic rigor.

- I assume that the balanced instance is $\tilde{I}$ instead of $I$ in the statements of the theorems.

- The paper seems to over claim the contributions in the introduction, since essentially the results hold for balanced instances, instead of all the instances in the online resource allocation problems. It would be helpful to clarify these, to avoid ambiguity.

- the concept of agents do not seem to appear in the general problem formulation. This necessitates further clarifications.

---

> ### Author Rebuttal · Authors · 2025-07-30
>
> We thank the reviewer for the careful review. Below, we address each question in detail.
>
> *Q: The paper could benefit from another round of polishing and it current form at is less ready for publication, since the current structure is a bit hard to follow with respect to the logic connections. For example, the introduction part focuses on heavily on the applications, but the applications mentioned in Section 5 do not match these applications exactly. One can get easily lost in the last two paragraphs before the Organization in Introduction. These two paragraphs do not help introducing this paper and its main theme. Also, merging the problem formulation with the contributions is somewhat confusing. Having a separate section of the problem formulation would make things clearer.*
>
> The last two paragraphs before the organization in the introduction are integral to the paper.
>
> The penultimate paragraph motivates the maximization problems (MaxMin, Nash Social Welfare, $p$‑means, etc.): it explains the “value to agents” viewpoint and asserts that our vector framework yields $(1-\varepsilon)$ guarantees there, contrasting with strong lower bounds without learning. This constitutes one half of the results in this paper, complementing the results for the minimization problems. Due to space limitations, the maximization proofs are in Appendix F. One of the two applications in Section 5, Nash Social Welfare, falls in the maximization category.
>
> The final paragraph describes the learnability core: we prove the existence of bounded, robust parameters—scaling with the number of coordinates and $\varepsilon$—and use this to obtain PAC learnability and stability to small estimation noise. Again, these are key contributions of the paper. The detailed proofs for learnability are in Appendix C.
>
> *Q: The proof of Lemma 4.4 is missing, or I may miss something. It would be helpful if the authors could point to the place.*
>
> After the statement of Lemma 4.4, the entire remaining part of Section 4 is dedicated to the proof of Lemma 4.4. The sentence “The remainder of this section gives a proof of Lemma 4.4” opens a proof sketch, and the proof is concluded at the end of the section (bottom of page 8, lines 311–314).
>
> *Q: MaxMin and MinMax are not defined with mathematic rigor.*
>
> MaxMin and MinMax are formally defined in lines 211–214. MinMax is the minimization of $f(v)=\max_{i\in[d]} v_i$, and MaxMin is the maximization of $f(v)=\min_{i\in[d]} v_i$.
>
> *Q: I assume that the balanced instance is $\tilde{I}$ instead of $I$ in the statements of the theorems.*
>
> We use the notation $I(\tilde{v},\tilde{k})$ for the transformed (balanced) instance, i.e., it is the instance with inputs $\tilde{v},\tilde{k}$.
>
> *Q: The paper seems to over claim the contributions in the introduction, since essentially the results hold for balanced instances, instead of all the instances in the online resource allocation problems. It would be helpful to clarify these, to avoid ambiguity.*
>
> We believe there is a misunderstanding of the result. We emphasize that our algorithm applies to all instances and not just balanced instances. The preprocessing that converts a general instance to a balanced instance is a part of our overall online algorithm. Conceptually, it removes options that should not be considered by the allocation. Subsequently, the algorithm for balanced instances is applied to the transformed instance and this constitutes the final solution for the original instance. Once again, we emphasize that the conversion to balanced instances is an algorithmic technique and not a restriction on the input, and the final solution and bounds are for the original instance, not the transformed (balanced) instance.
>
> In fact, this idea of preprocessing is also used in prior work such as [25]. To see why this is necessary, consider an instance of a makespan minimization problem where one option has $p_{i_1,j}=1$ and $p_{i_2,j}\to\infty$. Then, without preprocessing, we must have $\alpha_{i_1}/\alpha_{i_2}\to\infty$, making the parameter space unbounded.
>
> *Q: the concept of agents do not seem to appear in the general problem formulation. This necessitates further clarifications.*
>
> Our general formulation is stated in terms of resource coordinates. The familiar notion of agents is simply a special case: in maximization or fairness settings (e.g., Nash social welfare) we identify each coordinate with one agent, so the cumulative value of coordinate $i$ is exactly agent $i$’s utility. We already introduce this agent view in the Introduction and use it again in Section 5; in the revision we will add a brief sentence in the formal problem‑formulation section to make this mapping explicit and avoid any ambiguity.

---

> > ### Comment · Reviewer_NQoi · 2025-08-07
> >
> > I thank the authors for their responses. I believe that the writing of the paper could benefit from substantial changes/rewriting, as the current version is not easy to follow and the necessary modifications remain unclear based on the current rebuttal. I will increase my score, as some other comments are addressed accordingly.

---

> > > ### Author Response · Authors · 2025-08-07
> > >
> > > We thank the reviewer for the response to our rebuttal. We appreciate the more positive view of our work. We will make a thorough revision to polish the writing further, including highlighting why each part of the introduction is about an integral contribution of the paper.

---

### Official Review · Reviewer_ijy6 · 2025-07-03

**Clarity:** 3
**Significance:** 3
**Originality:** 3
**Rating:** 5
**Confidence:** 3

**Summary:**

This paper introduces a new, general framework for learning-augmented online allocation problems. Online allocation problems involve an algorithm making choices from a set of options at each step, which are represented as vectors, with the goal of minimizing or maximizing accumulated costs or rewards for a set of agents. This is represented by the value of an objective function on the sum of the vectors chosen at each step by the algorithm. Some problems that can be formalized in this way include machine scheduling, network routing (minimization), and fair allocation (maximization).


The proposed framework uses a single d-dimensional vector $\alpha$ of learned weights to produce a nearly optimal solution using an exponential assignment rule. The main result achieves a (1+$\epsilon$)-approximation for minimization or (1−$\epsilon$)-approximation for maximization for any $\epsilon>0$.  To that end, It leverages convex programming duality and a preprocessing step to convert the instance to a so-called “balanced” one. The authors also provide a modified algorithm to achieve a trade-off between consistency (performance with accurate predictions) and robustness (worst-case guarantees with unreliable predictions). The paper finally demonstrates the applicability of this framework to various problems in routing, scheduling, and fair allocation including online routing and online Nash Social Welfare maximization.


Minor comments:
I would suggest adding the relevant references to the first paragraph (e.g lines 24-25) of the introduction for clarity.

**Questions:**

1. Can you be more precise on the consistency robustness trade-off? Specifically, how is the confidence parameter $\lambda$ chosen and what is the dependence on the error of the learned parameter vector?  Is there an adaptive mechanism to set the confidence parameter dynamically based on the perceived accuracy of predictions, rather than a fixed value?

2. Could the convex programming formulation be adapted to handle different types of constraints or other settings beyond allocation problems?

3. For problems requiring integral solutions, how significant is the "small items assumption" in practice, and what are the implications if it doesn't hold perfectly?

**Ethical Concerns:**

["NO or VERY MINOR ethics concerns only"]

**Final Justification:**

The authors have answered my question in a satisfactory way. Therefore, I am keeping my recommendation for acceptance.

**Limitations:**

Yes

**Quality:**

3

**Strengths And Weaknesses:**

This paper's main strength is its development of a general learning-augmented algorithmic framework for online allocation problems, which can be widely applicable. The results leverage machine learned predictions to achieve almost optimal allocations with no prediction error while being robust to prediction error as well. In terms of techniques, the novel application of convex programming duality could also influence future research in the field. On the other hand, I would like to see more justification for the “small items assumption”, which is required both for learnability of the predictions and the rounding of the fractional solutions.

---

> ### Author Rebuttal · Authors · 2025-07-30
>
> We are glad the reviewer liked the paper and appreciate the insightful comments. Below, we address each question in detail.
>
> *Q: Can you be more precise on the consistency robustness trade-off? Specifically, how is the confidence parameter $\lambda$ chosen and what is the dependence on the error of the learned parameter vector? Is there an adaptive mechanism to set the confidence parameter dynamically based on the perceived accuracy of predictions, rather than a fixed value?*
>
> The use of a confidence parameter follows standard practice in learning-augmented algorithms, see e.g. [11], [31], and other papers such as
>
> *Gollapudi, S., & Panigrahi, D. (2019). Online algorithms for rent-or-buy with expert advice. ICML 2019.*
>
> As in prior work, the confidence parameter is application-dependent and typically tuned from data—for example, based on the number of prior samples or the discrepancy between predicted and actual performance on a validation window. It balances consistency (staying close to the learned policy) with robustness (falling back to a worst-case baseline). Fully automatic, near-optimal tuning is precluded by lower bounds without further assumptions. In practice, adaptive schemes work well: by tracking prediction error over time, the confidence level can be adjusted dynamically.
>
> *Q: Could the convex programming formulation be adapted to handle different types of constraints or other settings beyond allocation problems?*
>
> Yes, we believe our techniques can extend to broader classes of problems. Our max‑entropy template should extend cleanly to additional linear side constraints, and with appropriate smoothing or convex conjugates it may also handle certain separable concave terms (e.g., diminishing‑returns utilities). Formal guarantees would need a careful re‑derivation of the dual sensitivity bounds, so this is a good follow‑up direction for future work.
>
>
> *W: On the other hand, I would like to see more justification for the “small items assumption”, which is required both for learnability of the predictions and the rounding of the fractional solutions.*
>
> *Q: For problems requiring integral solutions, how significant is the "small items assumption" in practice, and what are the implications if it doesn't hold perfectly?*
>
> In real systems, “small items” are the norm or can be achieved with light batching. The atomic units—packets or flowlets in networks, requests or shards in services, impressions in online advertising—individually have small impact relative to total capacity. So, splitting traffic or budget across options already operates at fine granularity. Operators also aggregate over short time windows to smooth variability; this makes each decision step small enough that averages are stable, learning from samples is reliable, and randomized implementation (hashing, probabilistic routing, paced spending) stays close to targets.
>
> If the small items assumption doesn't hold perfectly, then the approximation factor of our algorithm degrades gracefully with increase in size of the largest item. Furthermore, when there’s a light tail of large items, those few can be handled separately, e.g. by reserving capacity, while the vast majority remain small, preserving both learnability and concentration in practice.
>
> Indeed, the small items assumption is quite standard in the literature for allocation problems, see e.g.,
>
> *Mehta, A., Saberi, A., Vazirani, U., & Vazirani, V. (2007). Adwords and generalized online matching. Journal of the ACM.*
>
> *Devanur, N. R. & Hayes, T. P. (2009). The adwords problem: online keyword matching with budgeted bidders under random permutations. EC 2009.*
>
> *Gollapudi, S., & Panigrahi, D. (2014). Fair Allocation in Online Markets. CIKM 2014.*
>
> *Agrawal, S., & Devanur, N. R. (2014). Fast algorithms for online stochastic convex programming. SODA 2014.*

---

### Official Review · Reviewer_e2Ps · 2025-07-06

**Clarity:** 3
**Significance:** 3
**Originality:** 3
**Rating:** 4
**Confidence:** 4

**Summary:**

This paper develops a unified learning-augmented framework for online allocation problems. The key contribution shows that a single learned parameter vector α enables an exponential assignment rule achieving (1±ε)-approximations for well-behaved objectives (routing, scheduling, fair allocation). Using convex programming duality and perturbation analysis, the authors prove existence of bounded, learnable parameters.

**Questions:**

Can you 1.provide concrete examples where fractional solutions are directly applicable without rounding? The paper's main guarantees are for fractional solutions, yet most applications (routing, scheduling) require integral decisions.

2.The three-step preprocessing appears quite aggressive, potentially changing the instance structure significantly. For example, provide an example where the algorithm fails without preprocessing? Also, it's helpful to quantify how much the preprocessing changes typical instances (what fraction of options are removed in Step 2)?

3.Provide the explicit algorithm for learning α from samples?

**Ethical Concerns:**

["NO or VERY MINOR ethics concerns only"]

**Final Justification:**

Issues Resolved:
Preprocessing misunderstanding: Clarified as algorithmic technique, not input restriction; guarantees apply to original instances
Fractional solutions: Provided compelling real-world applications (SDN, load balancing, ad pacing)
Learning framework: Distinguished analysis tool (convex program) from actual algorithm (ERM)

**Limitations:**

The authors partially address limitations throughout the paper but would benefit from a dedicated "Limitations" section consolidating:

Key restrictions: Fractional solutions only, required preprocessing that alters problem structure, i.i.d. assumption, and limitation to monotone/homogeneous objectives

Practical applicability gap: The theoretical unification is elegant but preprocessing and fractional nature limit direct deployment. The "learning" aspect is somewhat misleading—it's really about proving good fixed parameters exist rather than adaptive learning.

**Quality:**

3

**Strengths And Weaknesses:**

Strength:
Using convex duality to derive the exponential assignment rule from KKT conditions provides good theoretical grounding. The perturbation analysis showing dual variables can be bounded and discretized to poly(d,1/ε) is also well-executed.

Weakness:
The convex program requires knowing the entire instance, making the "learning" aspect somewhat artificial. The paper solves fractional problems, with integral solutions in the appendix. The (1±ε) guarantees are relative to fractional problems, not integral ones. There seems to be a mismatch of what the paper claims to contribute. The three-step balanced instance transformation also requires better motivation. It is essential to bound the exponential terms but deviate from reality it claims to model.

---

> ### Author Rebuttal · Authors · 2025-07-30
>
> We thank the reviewer for their careful reading of our manuscript and for the insightful feedback. Below, we address each question and weakness raised by the reviewer in detail.
>
> *Q: Can you provide concrete examples where fractional solutions are directly applicable without rounding? The paper's main guarantees are for fractional solutions, yet most applications (routing, scheduling) require integral decisions.*
>
> We give examples of applications where fractional solutions are directly applicable:
>
> - Traffic engineering / SDN: requests (or flows) are commonly split across multiple paths; fractional solutions translate directly to per‑path splitting ratios, see e.g.,
>
> *Salimifard, K., & Bigharaz, S. (2022). The multicommodity network flow problem: state of the art classification, applications, and solution methods.*
>
> *Alon, N., Awerbuch, B., Azar, Y., Buchbinder, N., & Naor, J. (2006). A general approach to online network optimization problems. ACM Transactions on Algorithms.*
>
>  - Cloud and datacenter load balancing: tasks/data shards are shared across machines or queues using probabilistic routing; fractional rates are implemented by hashing or randomized assignment, see e.g., [17], [23].
>
>  - Online advertising / budget pacing: budgets are split fractionally across channels/segments; the objective is smooth (Nash welfare or $\ell_p$) over aggregate spends, see e.g., [24]  and
>
> *Buchbinder, N., Jain, K., & Naor, J. (2007). Online primal-dual algorithms for maximizing ad-auctions revenue. In European Symposium on Algorithms.*
>
> Moreover, there has been prior work on learning-augmented algorithms that obtain fractional solutions, see e.g.,
>
> *Anand, K., Ge, R., Kumar, A., & Panigrahi, D. (2022). Online algorithms with multiple predictions. In International Conference on Machine Learning.*
>
>
> *W: The paper solves fractional problems, with integral solutions in the appendix. The (1±ε) guarantees are relative to fractional problems, not integral ones.*
>
> We have already discussed applications where fractional solutions can be directly applied. For applications that require integral solutions, we provide a rounding algorithm in Appendix D. We emphasize that after rounding, we get a $(1\pm \varepsilon)$ approximation for integral solutions as well under the standard small items assumption. In real systems, “small items” are typical or can be enforced through light batching. The atomic units—packets or flowlets in networks, requests or shards in services, impressions in online advertising—each have negligible impact relative to total capacity. Consequently, traffic or budget is already split across options at fine granularity. In practice, operators also aggregate decisions over short time windows to smooth variability; this keeps each step small enough that averages remain stable, sampling‑based learning is reliable, and randomized implementations (hashing, probabilistic routing, paced spending) track their targets closely. If a light tail of large items exists, those few can be handled explicitly—either by reserving capacity or by routing them individually—while the vast majority stay small, preserving both learnability and concentration guarantees.
>
> *Q: The three-step preprocessing appears quite aggressive, potentially changing the instance structure significantly. For example, provide an example where the algorithm fails without preprocessing? Also, it's helpful to quantify how much the preprocessing changes typical instances (what fraction of options are removed in Step 2)?*
>
> *W: There seems to be a mismatch of what the paper claims to contribute. The three-step balanced instance transformation also requires better motivation. It is essential to bound the exponential terms but deviate from reality it claims to model.*
>
> We believe there is a misunderstanding of the result. We emphasize that our algorithm applies to all instances and not just balanced instances. The preprocessing that converts a general instance to a balanced instance is a part of our overall online algorithm. Conceptually, it removes options that should not be considered by the allocation. Subsequently, the algorithm for balanced instances is applied to the transformed instance and this constitutes the final solution for the original instance. Once again, we emphasize that the conversion to balanced instances is an algorithmic technique and not a restriction on the input, and the final solution and bounds are for the original instance, not the transformed (balanced) instance.
>
> In fact, this idea of preprocessing is also used in prior work such as [25]. To see why this is necessary, consider an instance of a makespan minimization problem where one option has $p_{i_1,j}=1$ and $p_{i_2,j}\to\infty$. Then, without preprocessing, we must have $\alpha_{i_1}/\alpha_{i_2}\to\infty$, making the parameter space unbounded.
>
> *Q: Provide the explicit algorithm for learning α from samples?*
>
> In the appendix (lines 591–594), we present the learning algorithm: an ERM procedure that outputs the parameter values on the discrete net with minimum empirical loss.
>
> *W: The convex program requires knowing the entire instance, making the "learning" aspect somewhat artificial.*
>
> Finally, the convex program is used in the analysis to establish the existence of suitable parameters and to derive tight, coordinate-wise sensitivity bounds. It is not part of the online algorithm, which simply applies a fixed exponential rule using a learned vector $\alpha$. As shown in the paper, $\alpha$ can be estimated from data via finite-class empirical risk minimization (ERM) over a bounded net constructed from the sensitivity bounds. So, the learning is via the use of ERM on data as usual, but learnability of the parameters and the fact that the learned parameters yield a near-optimal solution are established via the convex program.

---

> > ### Comment · Reviewer_e2Ps · 2025-08-03
> > **Re: Rebuttal**
> >
> > Thanks for the clarification; the rebuttal successfully clarifies that preprocessing is an algorithmic technique (not an input restriction) and provides compelling real-world examples where fractional solutions are directly applicable. After careful consideration, I will revise the scores to reflect a more positive interpretation of the work's contribution.

---

> > > ### Author Response · Authors · 2025-08-04
> > > **Response to reviewer e2Ps**
> > >
> > > Dear reviewer e2Ps,
> > >
> > > Thank you for your positive assessment of the paper in light of the rebuttal. We really appreciate it!
> > >
> > > Best,
> > > Author(s)

---

### Official Review · Reviewer_cbL3 · 2025-07-15

**Clarity:** 3
**Significance:** 4
**Originality:** 3
**Rating:** 5
**Confidence:** 4

**Summary:**

The paper studies a general framework to solve online allocation problems via convex programming in the algorithms-with-predictions  model.  In the online algorithms with predictions framework, the goal is to leverage machine-learned predictions to make improved decisions when dealing with an online problem.  The goal is to achieve a better competitive ratio compared to a worst-case algorithm without any predictions.  The paper uses this model to provide an algorithm for a general online optimization problem such as online routing in which the algorithm is provided a prediction vector \alpha (with one parameter per edge) and goal is the minimize a global objective such as minimizing the max congestion one each edge.   Without predictions,  the best known competitive ratio for algorithms covered in this framework is logarithmic in the size of the input and the proposed learning-augmented algorithm achieves a (1+/- \eps) competitive ratio under prefect predictions.  To achieve robustness with respect to prediction errors, they propose a modified algorithm that essential reverts to the worst-case algorithm if the error accumulates.

Their framework, algorithm and results follow closely that of Cohen and Panigrahi [17] and generalize it to handle inputs that are vectors and use it solve problems that are not covered by [17] such as routing and fair allocation.


Minor typos:
Line 46 typo:  is logarithmic in the "number of size" of the the network [2]

**Questions:**

All my questions are about my only concern:  identifying the overlap with past work: citation [17] in the paper.  The following questions are trying to get more details about this.

1. What is the difference between the PAC learnability proof (Theorem 2.2) from the learnability bounds in [17]?
2. What is the main technical difference between the problems that cannot be handled by the framework in [17] vs the current paper?  The description that [17] cannot handle general vector needs explanation:  one way to interpret the divisible weights p_{i, j} in [17] is to view it as a d-dimensional vector (one per agent) that is received each time an item j appears.
3.  On line 177, the paper says in comparison to [17] "we simplify the previous approaches as well by requiring only a single variable per dimension":   it appears that [17] also requires one learned-parameter per dimension d so this needs further clarification.
4.  Is the fair allocation problem with Nash Social welfare objective already covered by the framework in [17]?  What does "matching previous results in [17]" mean on line 154 in the paper.

I am willing to raise my score to accept if this overlap is sufficiently addressed by the authors clarifying the new insights/contributions provided by this paper over [17].

**Ethical Concerns:**

["NO or VERY MINOR ethics concerns only"]

**Final Justification:**

I am raising my score as the authors response adequately address my questions and concerns.

**Quality:**

4

**Strengths And Weaknesses:**

Strengths:
-  The paper provides a way to solve general online convex optimization problems in the algorithms-with-predictions model.
-  Applying their general framework to online nash social welfare with predictions improves upon the prior best-known competitive ratio from logarithmic to (1+/eps).
-  They provide the first learning-augmented algorithm for the online routing problem which is a fundamental optimization problem
-  The paper shows that the predictions are PAC learnable, as well as provide new techniques such as perturbation analysis which are of independent interest to the community.

Weakness:
 - The main weakness of the paper is that it does adequately address the overlap with prior work Cohen and Panigrahi [17] in the current write up.  The framework and algorithm borrow significantly from [17] and the new ideas and insights are not clearly addressed.   See questions below.

---

> ### Author Rebuttal · Authors · 2025-07-30
>
> We are glad that the reviewer liked the paper and appreciate the insightful comments. Below, we address each question in detail and conclude with a concise summary of the main differences between our work and [17] as requested by the reviewer.
>
> *Q: What is the difference between the PAC learnability proof (Theorem 2.2) from the learnability bounds in [17]?*
>
> Both works establish PAC learnability by restricting to a finite hypothesis class and applying standard finite‑class generalization bounds. The crux in both settings is to bound the parameter space. In prior work [17, 22, 25], a simple rescaling of the learned parameters yields an equivalent (or near‑equivalent) allocation, so one can cap parameter magnitudes with only a negligible loss. In our vector setting, this rescaling trick fails. Instead, we bound the parameters via a max‑entropy dual and a coordinate‑wise sensitivity analysis.
>
> *Q: What is the main technical difference between the problems that cannot be handled by the framework in [17] vs the current paper? The description that [17] cannot handle general vector needs explanation: one way to interpret the divisible weights p_{i, j} in [17] is to view it as a d-dimensional vector (one per agent) that is received each time an item j appears.*
>
> In [17], for each arriving item $j$, each option assigns it to a single agent $i$ with value/cost $p_{i,j}$. Our paper is more general. Here, for each arriving item $j$, each option $k$ corresponds to an entire vector $v_{j,k}\in\mathbb{R}^d_{\ge 0}$ that adds value/cost for multiple agents simultaneously. This captures coupled, multi‑coordinate effects within a single choice. So, [17] is a special case of our setting where each vector has a single non-zero entry. In comparison, we allow “general vectors”.
> Such a coupled choice is necessary in applications like online routing. Here, each option is a path which adds load on each edge of the path. This can be modeled in our framework by encoding edges as coordinates. For example, suppose there are three edges $e_1,e_2,e_3$ and the feasible path vectors (i.e., the options) are $v_1=(1,1,0)$ and $v_2=(0,0,1)$. (So, one path comprises the first two edges and the second path comprises only the third edge.) Our model can choose between $v_1, v_2$ or a convex combination of these two options. For instance, if one chooses fraction $\tau$ for the first option and fraction $1-\tau$ for the second option, then the load on the edges increases by $(\tau,\tau,1-\tau)$. The model in [17] can only capture the situation where every path is a single edge. If a path has multiple edges, then the options correspond to vectors with multiple non-zero entries, which cannot be captured by the model in [17].
> In terms of techniques, [17] uses a form of proportional allocation. Unfortunately, proportional allocation cannot handle coupled, multi-coordinate effects due to general vectors in our setting.
>
> *Q: On line 177, the paper says in comparison to [17] "we simplify the previous approaches as well by requiring only a single variable per dimension": it appears that [17] also requires one learned-parameter per dimension d so this needs further clarification.*
>
> In our framework, there is a single learned vector over resource coordinates which is also there in [17], but we do not have any global exponent or auxiliary scaling vector which are both present in [17].
>
> *Q: Is the fair allocation problem with Nash Social welfare objective already covered by the framework in [17]? What does "matching previous results in [17]" mean on line 154 in the paper.*
>
> [17] provides a near‑optimal fractional guarantee for online Nash Social Welfare (NSW) where every option allocates the item to a single agent and adds value to the agent. Our result matches the bounds for this case, but we also generalize it further to options having vector utilities, i.e.,  where an option can add (possibly different amounts of) value to multiple agents simultaneously. In other words, choosing an option adds a utility vector to the agents. This coupled, multi‑beneficiary setting is captured by our vector framework but is not handled in [17]. We will clarify this in the paper.
>
>
> In summary, the following are the key differences between our paper and [17]:
>
> 1. Scope of the model: [17] is an assignment model in which each item has to be split across agents; our framework handles general vector options that update many coordinates jointly, enabling coupled applications such as online routing and vector‑utility NSW that [17] does not encode.
>
> 2. Parameterization: We use a single learned vector over resource coordinates without any auxiliary global exponent or per‑agent scaling, both of the latter being present in [17].
>
> 3. Techniques: The rescaling trick used in [17, 22, 25] does not work for vectors; we instead derive parameters from a max‑entropy dual with a coordinate‑wise sensitivity bound giving an explicit box $0 \le \alpha_i \le \tfrac{d^2}{\varepsilon}\log(d/\varepsilon)$, which yields a robust finite net and PAC guarantees. Indeed, the connection that we draw to the max-entropy convex program and the subsequent use of convex programming duality is a novel contribution that we believe does not appear in this line of work prior to our paper.
>
> To highlight these differences, we will add a focused comparison paragraph and a routing example to make the distinction clear.

---

### Note · Authors · 2025-08-11

We thank the reviewers for the insightful reviews and follow up questions post rebuttal. We summarize a few of the main points from the rebuttal and subsequent discussions
- our results hold for all instances and not just balanced instances
- our paper is substantially different from [17] both in scope/applications (e.g., routing and multi-item NSW maximization) and techniques/parametrization (convex programming duality, single learned vector with no additional global exponent)
- our results hold for fractional and integral solutions, the latter under the standard small items assumption
- knowledge of the convex program is not required either by the learning algorithm (ERM) or the online algorithm (exponential rule); it is only used in the existential analysis
- our paper contains full proofs of all stated lemmas and theorems, and the topics covered in the introduction, including in the last two paragraphs, are expanded further in separate sections in the main body or the appendix

We thank the reviewers for suggestions about writing, which we will use to improve clarity of presentation.

---

### Decision · Program_Chairs · 2025-09-17

**Decision:**

Accept (poster)

**Comment:**

The submitted paper introduces a learning-augmented algorithmic framework for online allocation problems, aiming to achieve near-optimal solutions using a single-dimensional vector of learned weights. The framework applies to a wide range of combinatorial optimization problems, including routing, scheduling, and fair allocation for welfare maximization. The approach leverages convex programming duality as a key tool.

**Strengths**
* Novelty: The paper provides a way to solve general online convex optimization problems in the algorithms-with-predictions model. Applying their general framework to online Nash social welfare with predictions improves upon the prior best-known competitive ratio from logarithmic to (1+/eps).
* Practical applicability and relevance: The framework is applicable to a wide range of relevant problems, including routing, scheduling, and fair allocation, with real-world examples provided for fractional solutions (e.g., traffic engineering, cloud load balancing, and online advertising).
* Robustness and consistency: The proposed algorithm achieves a trade-off between robustness (worst-case guarantees) and consistency (performance under accurate predictions).


**Weaknesses**
* Clarity and presentation: Several reviewers noted that the paper's structure and writing could be improved. The introduction, in particular, was criticized for being hard to follow, with some sections lacking clear connections. Also the necessity of preprocessing to handle balanced instances was initially misunderstood as a restriction on the input, which could have been better explained in the main text.
* Overlap with prior work: While the authors clarified the distinctions from Cohen and Panigrahi [17], the initial write-up did not sufficiently highlight these differences, leading to confusion among reviewers. This distinction must be made clear in a revised version of the paper.
* Assumptions: The "small items assumption" for obtaining integral solutions was questioned, though the authors provided satisfactory justifications in the rebuttal (which should be added to the final paper).

**Discussion**
The following key points were discussed:
* Overlap with [17]: The authors clarified that their framework generalizes [17] by handling vector-based inputs and coupled effects, which are essential for applications like online routing. They also highlighted novel techniques, such as the use of max-entropy duals and coordinate-wise sensitivity analysis.
* Fractional vs. integral solutions: The authors provided real-world examples where fractional solutions are directly applicable and explained how integral solutions can be obtained with a small approximation loss under standard assumptions.
* Clarity Improvements: The authors committed to revising the paper to improve clarity, including restructuring the introduction and explicitly defining key concepts like agents and balanced instances.

**Recommendation**
While the paper has some weaknesses in presentation and clarity (which can be easily resolved), its technical contributions were unanimously appreciated by the reviewers. The authors have also addressed most of the other reviewers’ concerns convincingly. The framework's generality, theoretical rigor, and practical relevance make it a valuable contribution to the field. Hence, in line with the clear majority of the reviewers, I am recommending the acceptance of the paper.